# Inborn Errors of Metabolism with Ataxia: Current and Future Treatment Options

**DOI:** 10.3390/cells12182314

**Published:** 2023-09-19

**Authors:** Tatiana Bremova-Ertl, Jan Hofmann, Janine Stucki, Anja Vossenkaul, Matthias Gautschi

**Affiliations:** 1Department of Neurology, University Hospital Bern (Inselspital) and University of Bern, 3010 Bern, Switzerland; jan.hofmann1@students.unibe.ch (J.H.); janine.stucki@students.unibe.ch (J.S.); 2Center for Rare Diseases, University Hospital Bern (Inselspital) and University of Bern, 3010 Bern, Switzerland; 3Division of Pediatric Endocrinology, Diabetes and Metabolism, Department of Paediatrics, Inselspital, Bern University Hospital, University of Bern, 3010 Bern, Switzerland; anjamaria.vossenkaul@insel.ch (A.V.); matthias.gautschi@insel.ch (M.G.); 4Institute of Clinical Chemistry, Inselspital, Bern University Hospital, University of Bern, 3010 Bern, Switzerland

**Keywords:** hereditary metabolic ataxia, cerebellar ataxia, inborn error of metabolism, symptomatic treatment, disease-modifying treatment, GM2-gangliosidosis, Niemann-Pick type C, metabolic diseases, lysosomal storage disorders, abetalipoproteinemia, cerebrotendinous xanthomatosis, disorders of carbohydrate metabolism, neurodegeneration, neuroprotection, ocular motor, biomarkers, acetyl-L-leucine, acetyl-DL-leucine

## Abstract

A number of hereditary ataxias are caused by inborn errors of metabolism (IEM), most of which are highly heterogeneous in their clinical presentation. Prompt diagnosis is important because disease-specific therapies may be available. In this review, we offer a comprehensive overview of metabolic ataxias summarized by disease, highlighting novel clinical trials and emerging therapies with a particular emphasis on first-in-human gene therapies. We present disease-specific treatments if they exist and review the current evidence for symptomatic treatments of these highly heterogeneous diseases (where cerebellar ataxia is part of their phenotype) that aim to improve the disease burden and enhance quality of life. In general, a multimodal and holistic approach to the treatment of cerebellar ataxia, irrespective of etiology, is necessary to offer the best medical care. Physical therapy and speech and occupational therapy are obligatory. Genetic counseling is essential for making informed decisions about family planning.

## 1. Introduction

Hereditary (or inherited) cerebellar ataxias are a heterogeneous group of individually rare, progressive, and neurological disorders that affect the cerebellum and the cerebellar network, the parts of the brain responsible for coordinating movement and balance. Ataxias are characterized by an unsteady gait leading to recurrent falls, clumsiness, difficulty with coordination, slurred speech, and impaired fine motor skills. Neuro-ophthalmological and -otological manifestations can also be a component of and/or a factor leading to ataxia given the common infratentorial pathways’ and networks’ dysfunction. Ataxias can be caused by inborn errors of metabolism (IEM) which have mostly autosomal recessive, but sometimes X-linked or mitochondrial, inheritance.

Considering that the impaired metabolic pathways in IEM have an impact on numerous functions, cerebellar ataxia is only one component of the clinical picture, even though it may be the leading symptom. Vulnerabilities of specific neuronal populations in the central nervous system (CNS) can lead to disruptions in functional networks causing specific but often multifaceted patterns of neurological deficits along with psychiatric and systemic manifestations. Ataxia is the most common movement disorder in IEM. It occurs in various combinations with other movement disorders. For example, defects of intermediary metabolism are often accompanied by tremors, whereas IEM of complex molecules are more often present with dystonia. Based on the temporal pattern of the occurrence of symptoms, we can distinguish between intermittent metabolic ataxias, occurring due to the worsening of the underlying metabolic abnormality (i.e., metabolic decompensation, commonly occurring in cases of IEM of the intermediary metabolism), and the chronic static or progressive metabolic ataxias, which are usually caused by deficiencies of enzymes or transporters involved in the synthesis or breakdown of complex molecules, leading to chronic brain cell dysfunction and neurodegeneration, e.g., by intracellular storage. Ataxia, as a leading symptom, often occurs in milder defects, including adult-onset forms of IEM [1].

Currently, there is no cure for hereditary cerebellar ataxias, representing a significant unmet medical need.

Thus, it is of paramount importance to promptly diagnose an underlying IEM in order to prevent potentially irreversible damage to the CNS caused by metabolic decompensations, which often occur in disorders of intermediary metabolism. Currently, the general therapeutic approach aims to manage symptoms, prevent or slow down disease progression (by introducing disease-specific treatments, if available), and improve quality of life. Recently, promising gene therapy trials have been initiated for some of the IEM, e.g., first-in-human [2] trials for GM2-Gangliosidoses or metachromatic leukodystrophy [3,4], in the latter showing a remarkable long-term improvement in neurological state compared to natural history data. Preliminary experimental gene therapies for some IEM have been tested in animal studies [5]. Some IEM have a very specific pathomechanism. Thus, targeted treatments may lead to either stabilization/slowing down of the disease progression or to diminished signs and symptoms of the disease. A few general principles with regard to IEM can be formulated:The earlier the disease starts, the more severe it generally is, and the severity of the disease often correlates with the residual activity of the affected gene product (e.g., enzyme);Emergency treatment for acute decompensations of IEM of the intermediary metabolism generally includes a non-specific anticatabolic treatment with intravenous administration of high glucose with insulin, if needed, as well as temporary reduction or stopping of nutrition combined with more specific detoxification measures as indicated, such as ammonia scavengers in case of hyperammonemia (see disease-group-specific guidelines);Regarding the general symptomatic therapeutic principles, anti-seizure medication, together with antipsychotics, antidepressants, and muscle-relaxing agents, such as baclofen or tizanidine or in case of focal/segmental spasticity/dystonia treatment with botulotoxin should be used to relieve the burden of multiple disease symptoms.

Some metabolic ataxias can be diagnosed by biochemical screenings, either as part of a standardized newborn screening or once the neurological abnormalities are first noticed. For IEM without biochemical markers, first-line diagnosis is usually performed by genetic testing, which may be impeded by the absence of variants, the presence of variants of unknown significance, or other inconclusive findings. Functional investigations remain important in the work-up of undiagnosed patients. For example, pyruvate dehydrogenase deficiency can be corroborated by relatively simple biochemical assays on cultured fibroblasts or clinically with a glucose loading test.

Here, we review the most important grouped or individual IEM that may present with ataxia as the leading or accompanying disease manifestation. We describe the clinical picture and the current and potential future treatment options with a focus on lysosomal storage disorders. Table 1 provides a summary of the main points.

## 2. Disorders of Amino Acid Metabolism

### 2.1. Organic Acidurias

The organic acidurias, the propionic (PA), methylmalonic (MMA) and isovaleric (IVA) acidemias are caused by defects in the catabolism of branched-chain amino acids (BCAA). They belong to the most common causes of intermittent ataxias. Organic acidurias belong to the intoxication type of IEM. Untreated, they can cause brain damage in many ways, often through impairing brain energy metabolism [22]. Along with episodes of ataxia, symptoms include developmental delay and cognitive impairment, seizures, episodic vomiting, and lethargy. Other organic acidurias can cause chronic ataxia, e.g., mevalonic aciduria can present as chronic progressive ataxia, whereas non-progressive metabolic ataxia may be due to SSADH (succinic semialdehyde dehydrogenase) deficiency or L-2-hydroxy-glutaric aciduria.

#### Isovaleric (ORPHA:33 and OMIM: 243500), Propionic (ORPHA:35 and OMIM: 606054), and Methylmalonic Acidemias (ORPHA:289916 and OMIM: 606054)

Isovaleric Acidemia (IVA) is an autosomal recessive disorder of leucine catabolism. It is caused by mutations in the isovaleryl-CoA dehydrogenase (IVD) gene, leading to an accumulation of derivatives of isovaleryl-CoA, including isovaleryl (C5)-carnitine, which is assessed as part of the newborn screening in some countries. IVA can cause delayed psychomotor and cognitive development in some cases [23]. Only a subset of patients seem to suffer from neurological and/or cognitive symptoms [23]. Infants with IVA can develop episodes of metabolic acidosis, which are rather rare after nine years of age [24]. There are three phenotypes of IVA: (1) the symptomatic “acute neonatal” form typically presents with the first acute catabolic episode at less than a week of age, with potentially life-threatening metabolic acidosis, a characteristic odour of “sweaty feet”, lethargy, dehydration, vomiting and sometimes impaired consciousness and seizures; (2) the symptomatic “chronic intermittent” form presenting later in childhood, and (3) the mostly asymptomatic “mild” form, caused by a common missense mutation.

PA and MMA are due to defects in the breakdown of isoleucine and valine. Very similar to IVA, they present as either acute neonatal, chronic progressive or late-onset intermittent forms. Diagnosis by urinary organic acid analysis and genetic confirmation, as well as treatments are analogous to IVA as well.

*Therapy:* Protein-restricted diet, specific amino-acid supplementation, vitamin and oligoelement supplementation. Medication includes L-carnitine, +/− glycine in the case of IVA, or +/− sodium benzoate, and intermittent antibiotics in the case of PA and MMA. Avoidance of catabolic states, including emergency treatment in case of intercurrent illness and other situations with risk of decompensation is highly necessary. Consensus guidelines for the diagnosis, treatment and follow-up have been published [25].

### 2.2. Maple Syrup Urine Disease (ORPHA:268145 and OMIM: 248600)

Maple syrup urine disease is an autosomal recessive disorder of the branched-chain α-ketoacid dehydrogenase (BCKAD) complex, caused by mutations in the *BCKDHA*, *BCKDHB*, *DBT*, or *DLD* genes, which code for its catalytic subunits. These mutations result in elevated branched-chain amino acids (BCAA) in plasma, α-ketoacids in urine, and increased production of the pathognomonic disease marker, alloisoleucine [26]. Urine and other body fluids have a characteristic, maple syrup-like odour. Three distinct clinical phenotypes are described. Classic MSUD present soon after birth and is severe and potentially fatal. The intermediate and intermittent forms can be overlapping, presenting at any time of life, with stress- or infection-related decompensations, however adult-onset is rather rare. The clinical presentation of MSUD depends on the BCKAD residual activity, the leucine tolerance and the metabolic response to illness. Movement disorders, such as characteristic “bicycling” movements in neonates, dystonia, spasticity, choreoathethosis, however also cerebellar ataxia may be present, along with cognitive impairment, hyperactivity, sleep disturbance and hallucinations [26]. Ocular motor impairment can occur, varying from up-gaze supranuclear palsy and adduction deficit to absence of voluntary eye movements with absent vestibulo-ocular reflexes [27,28].

*Therapy:* Low-protein diet with restriction of leucine, isoleucine, and valine. Thiamine supplementation in thiamine-responsive MSUD (pathogenic variants in *DBT*). Phenylbutyrate acts on the kinase of the BCKAD and could become an additional treatment option in MSUD [29] (see also NCT01529060). During periods of protein catabolism, such as fever, infections, exercise, trauma, or surgery, individuals with MSUD can develop neurological deterioration, which may manifest as intermittent ataxia, due to acute leucine intoxication. Thus, emergency management is essential. Patients with severe disease may need liver transplantation.

### 2.3. Hartnup Disease (ORPHA:2116 and OMIM: 234500)

Hartnup disease is caused by a deficiency of the sodium-dependent neutral amino acid transporter in the proximal kidney tubules and jejunum, resulting from a mutation in the solute carrier family 6 member 19 (SLC6A19) gene [30,31]. Biochemically, Hartnup disorder is diagnosed via a corresponding amino acid excretion pattern [32]. The ensuing tryptophan deficiency leads to secondary niacin and serotonin deficiencies, thought to be responsible for the pellagra-like skin and the neurological manifestations, including ataxia and neuropsychiatric symptoms. However, most patients are asymptomatic (i.e., low penetrance).

*Therapy:* Supplementation with niacin (nicotinamide) or tryptophan-rich diet in order to correct the secondary deficiency. Symptomatic patients should strictly protect themselves from sun exposure and avoid photosensitizing drugs.

## 3. Lysosomal Storage Diseases

Disorders of lysosomes (LSD) and lysosome-related organelles (LROs) can be subclassified according to the biochemical type of stored material (e.g., into sphingolipidoses, mucopolysaccharidoses, glycoproteinoses, and neutral lipid storage disorders), the integral membrane proteins, post-translational modification, or lipofuscin accumulation.

### 3.1. Gaucher Disease (ORPHA:355)

Gaucher disease (GD) is a sphingolipidosis caused by mutations in the GBA gene, which encodes the enzyme glucocerebrosidase, leading to accumulation of glucosylceramide and its deacylated form, glucosylsphingosine. Even though there is a continuum of signs and symptoms in reality, three phenotypes have been distinguished historically: non-neuronopathic (GD1), acute neuronopathic with infantile onset (GD2), and chronic neuronopathic (GD3). In GD2, early signs in affected children are abnormal eye movements, including horizontal gaze palsy and slow saccades, saccadic initiation problems, and strabismus. Furthermore, early slowing of development followed by regression and severe developmental retardation, as well as visceral signs with hepatosplenomegaly, are prominent. GD3 has a later-onset and commonly presents with cerebellar ataxia and dystonia. Patients may develop (myoclonus) epilepsy that might be severe and lead to neurological deterioration. Patients also show a horizontal supranuclear saccade or complete gaze palsy that is compensated for by head movements, so-called head thrusts, indicating the brainstem involvement [33]. Vestibulo-ocular responses are also impaired [33,34].

*Therapy:* Intravenous enzyme replacement therapy (ERT) with recombinant glucocerebrosidase is available for visceral manifestations in all forms. However, its use in the severe GD2 form is considered futile. Eliglustat is the first-line substrate reduction therapy (SRT) in adult GD1 patients, showing comparable efficacy to ERT [35]. Prior to the use of eliglustat [36], the pharmacogenomics of CYP2D6, the primary metabolizing enzyme of eliglustat needs to be clarified in all patients in order to determine the adequate dosing. Miglustat is a second-line alternative SRT for adult patients. The potential advantage of these small molecules is their penetrance into the CNS, in contrast to ERT. Various investigational compounds directed at the CNS (including other SRT, arimoclomol, ambroxol, and others) are currently evaluated in phase 1–3 clinical trials. Arimoclomol [37] is thought to act by enhancing the endogenous chaperone system via the HSP70 pathway. Ambroxol [38], an over-the-counter sold mucolytic drug, acts as a pharmacological chaperone. Both these drugs penetrate easily into the central nervous system and could therefore potentially improve cerebral complications of the disease. In addition, an AAV-based gene therapy, so far only for the visceral form (GD1), is currently recruiting (NCT05324943).

### 3.2. GM2-Gangliosidoses (Tay–Sachs and Sandhoff Disease) (ORPHA:845 and OMIM: 272800)

The GM2-gangliosidoses (vGM2), namely Tay–Sachs disease (TSD), Sandhoff disease (SD), and AB variant, are caused by mutations in the *HEXA*, *HEXB,* or *GM2A* genes, respectively, which code for the α- or β-subunits of β-hexosaminidase or the GM2 activator protein. They are clinically indistinguishable from one another (but certain subtle visceral and skeletal features only occur in SD) [39]. The GM2-gangliosidoses can be divided into infantile-, juvenile- and adult-onset forms. Phenotypically, juvenile- and adult-onset forms feature cerebellar ataxia combined with the involvement of a second motoneuron, along with slowly developing intellectual disability and psychiatric manifestations including bipolar disorder or psychosis. The diagnostic standard is an enzymatic assay of β-hexosaminidase activities (usually carried out in serum and/or peripheral white blood cells) followed by genetic confirmation. In a mouse model of SD disease, a recombinant AAV-α or AAV-β was injected, which led to improved survival of these mice only if the treatment was initiated pre-symptomatically or very early in the disease manifestations [40].

*Therapy:* Currently, there is no approved treatment for GM2. *N*-acetyl-*L*-leucine (ALL) has shown symptomatic benefits in motor domains, quality of life in a positive rater-blinded clinical trial, and in a case report [21,41]. Additionally, a disease-modifying, neuroprotective effect was demonstrated in animal models [42,43]. A first-in-human study of TSHA-101 gene therapy for the treatment of infantile-onset GM2 is ongoing in Kingston, Ontario, Kanada (NCT04798235) [2], where three infants receive the AAV9 viral vector containing *HEXA* and *HEXB* genes intrathecally.

### 3.3. Metachromatic Leukodystrophy (ORPHA:512)

Metachromatic leukodystrophy (MLD) is caused by mutations in the *ARSA* gene, leading to a deficiency of the enzyme arylsulfatase A (ARSA). Rarely, it can be related to mutations in the prosaposin gene *PSAP*, leading to a deficiency of ARSA activator protein saposin B (SAP-B) [44] that normally stimulates the degradation of sulfatides by ARSA. ARSA deficiency leads to sulfatide storage in microglia, oligodendrocytes, and Schwann cells [45], leading to demyelination because sulfatides are the major component of the myelin membrane. Neuronal sulfatide storage also leads to neurodegeneration and neuroinflammation. The early-onset MLD form is characterized by rapid progression leading to death within the first decade of life. The late-infantile form is the most severe as all patients are tetraplegic and in a vegetative state by the end of the second year [46]. The juvenile-onset disease is heterogeneous, manifesting with cerebellar ataxia, gait disturbance, upper motor neuron signs, cognitive disability, behavioral difficulties, and peripheral neuropathy [47]. If seizures occur, the disease is fatal within six years of onset. A milder form or MLD, usually adult-onset, presents with cognitive decline as the first symptom [48]. It is characterized by dementia, behavioral changes, and psychiatric manifestations [49]. The typical brain MRI shows a tigroid (striped) pattern of white matter T2-weighted hyperintensities in ~70% of cases, over time accompanied by cortical atrophy [24].

*Therapy:* Allogenic hematopoietic stem cell transplantation (HSCT) is the first-line treatment option in patients with no or early MLD disease, treating CNS manifestations but not peripheral neuropathy [24,25,26]. Ex vivo gene therapy with HSCT (atidarsagene autotemcel or arsa-cel) [50] for early-onset MLD has been facilitated and approved by the European Medicines Agency (EMA) in December 2020 [51]. A lentiviral vector is used to transfer a functional *ARSA* gene into autologous HSCs [3]. The gene-corrected HSCs (arsa-cel) are then intravenously transferred back into patients, enabling engraftment of the transduced cells with elevated levels of ARSA enzyme activity. The two-years long-term trial [4], including 26 surviving children with pre- or early symptomatic late-infantile or early-juvenile MLD, showed improved ambulation and either normal development or stabilization of motor skills overall. Their cognitive development was normal with absent or postponed brain white matter and cortical involvement. Patients with late-infantile onset showed improved peripheral nerve conduction velocity. The benefits of arsa-cel treatment were most apparent in patients who were pre-symptomatic when treatment started. A phase I/II, open-label, and monocentric study of direct intracranial administration of a replication-deficient adeno-associated virus gene transfer vector serotype rh.10 expressing the human ARSA cDNA to children with MLD is ongoing (NCT01801709) [52].

### 3.4. Neuronal Ceroid Lipofuscinoses (NCLs) (ORPHA:216)

The neuronal ceroid lipofuscinoses (NCLs) are LSDs characterized by progressive degeneration of the brain and frequently of the retina with intracellular accumulation of autofluorescent lipopigment [53]. NCL genes are expressed in various tissues but the disease manifests mainly as CNS dysfunction. Early affected brain structures are the thalamus and cerebellum [54]. Five of the NCL variants are caused by defects in soluble lysosomal proteins: CLN1 disease due to palmitoyl protein thioesterase-1 (PPT1) deficiency, CLN2 disease due to tripeptidyl peptidase-1 (TPP1) deficiency, CLN5 disease due to CLN5 protein deficiency, CLN10 disease due to cathepsin D deficiency, and CLN13 due to cathepsin F deficiency. All other NCL variants are due to defects in lysosomal endoplasmatic reticulum or cytoplasmic vesicular transmembrane proteins [55]. Genetically, at least 14 different forms of NCL have been found and assigned a ”CLNx” number. All NCLs, except CLN4 disease, are recessive disorders. Individually, NCLs are rare conditions, yet collectively, they are the most frequent childhood neurodegenerative condition [53]. Clinical manifestations include vision loss, dementia, epilepsy, movement disorders, and storage of ceroid lipofuscin in neurons. NCLs differ in age of onset, clinical features, and rate of progression [56]. In CLN1 disease, there is a rapid, prematurely fatal regression between 6 and 24 months old, characterized by seizures, myoclonus, ataxia, and visual failure, followed by severe spasticity and a decreased level of consciousness [57]. The adult-onset form begins with cognitive decline and depression, with cerebellar ataxia, parkinsonism, and vision loss setting in later [58]. CLN2 disease is associated with the classic late infantile-onset form of NCL [59], with onset between two and four years of age. Typical symptoms include a speech delay, developmental plateau, epilepsy, polymorphic seizures, and over time ataxia, refractory nonepileptic myoclonus, and spastic quadriparesis. Similar to CLN1 disease, children with CLN2 onset past four years of age have a milder course with more prominent ataxia and less epilepsy [60]. CLN3 is a juvenile-onset form, encompassing vision loss, cognitive decline, behavioral problems, epilepsy, and parkinsonism [61]. Stuttering dysarthria is characteristic and develops usually after ten years of age. Ataxia is also a symptom of CLN4, CLN5, CLN6, CLN10, and CLN11. Myoclonus can be observed in CLN4 and CLN7 as well. CLN12, also known as Kufor-Rakeb syndrome (PARK9), is caused by loss-of-function variants in the predominantly neuronal P-type ATPase gene, ATP13A2. CLN12 presents with parkinsonian features of a mask-like face, rigidity, and bradykinesia but without tremor. Spasticity, supranuclear upgaze palsy, and dementia are symptoms of most affected patients [62]. The preferred diagnostic method is genetic testing with the possible addition of brain imaging, which can show cerebellar atrophy.

*Therapy:* In general, supportive care paired with pharmacological strategies. For epilepsy, antiseizure medications (such as valproate, lamotrigine, topiramate, and levetiracetam) are administered although epilepsy due to NCL is often resistant to treatment [63]. Baclofen can help against spasticity but movement disorders in NCLs can be difficult to treat because of the complexity of movement disorder combinations and the treatment-refractory symptoms. Disease-modifying therapy for CLN2 for patients aged three and older is recombinant human cerliponase alfa, the proenzyme form of human TPP1, and the enzyme deficient in CLN2, administered into the cerebrospinal fluid via the lateral ventricles [64].

### 3.5. Niemann-Pick Type A and B (ORPHA:618899)

Niemann-Pick types A and B are caused by deficient activity of the enzyme acid sphingomyelinase (ASM) [65]. All patients with types A and B Niemann-Pick disease have mutations in the gene encoding ASM (*SMPD1*) and thus the disease is more accurately referred to as ASM deficiency (ASMD). Patients suffering from type A ASMD present with failure to thrive in the neonatal period, hypotonia, and failure to reach developmental milestones. They rarely live past two or three years of age. Patients with type B have hepatosplenomegaly and pathologic alterations of their lungs but usually little to no neurological involvement. In patients with type B, the age of onset and rate of disease progression varies and they frequently live into adulthood. Serum triglycerides and LDL-cholesterol are commonly elevated, while HDL-cholesterol is low. Elevated oxysterols may provide an important clue for diagnosis (especially C-triol), even though somewhat less consistent than in type C ([66] and see below). Variant phenotypes feature cerebellar ataxia and neurodevelopmental delay leading to learning difficulties with more severe systemic manifestations [67,68]. Patients carrying the Q294K variant have an increased risk of false negative diagnosis. Pseudo-normal results are obtained from ASM assays employing fluorometric substrates, leading to substantial under-diagnosis in patients with the Q294K mutation [69]. Therefore, best practice guidelines recommend the use of the tandem mass spectrometry ASM assay, which provides a reliable diagnosis in Q294K-dependent cases.

*Therapy:* Recombinant human ASM (olipudase alfa) [70,71], based on the results of phase 1 and within-patient dose escalation phase 1b clinical trials, is approved by both the US Food and Drug Administration (FDA) and the European Medicines Agency (EMA). Its approval in Switzerland is underway (Xenpozyme^®^). In children with chronic ASMD, olipudase alfa was well-tolerated and showed significant, comprehensive improvements in disease pathology across a range of clinically relevant endpoints [72]. This finding was confirmed in the phase 2/3, randomized, double-blinded, placebo-controlled, and repeat-dose study (NCT02004691) [73].

### 3.6. Niemann-Pick Type C (ORPHA:646)

Niemann-Pick disease type C (NPC) is caused by mutations in the *NPC1* or *NPC2* genes, leading to an impaired intracellular lipid transport with its storage in lysosomes and late endosomes [74,75]. The accumulated cholesterol undergoes non-enzymatic oxidation, leading to the formation of oxysterols [76]. Oxysterols are used as blood-based biomarkers of NPC, together with certain plasma bile acids [77] (e.g., 3β,5α,6β-trihydroxycholanic acid) and certain lysosphingolipids [78] (e.g., lyso-SM-509), replacing filipin staining as first-line diagnostics.

NPC disease is highly heterogeneous, comprising systemic, neurological, and psychiatric signs [79]. Early- and late-infantile forms of NPC are present with severe hepatosplenomegaly, cataplexy, epilepsy, and a regress of already-gained neurological skills. These forms are rapidly progressive and prematurely fatal. In comparison, the juvenile and adolescent forms of NPC progress more slowly and present with cerebellar ataxia, dysarthrophonia, dystonia, cognitive decline, psychiatric involvement, vertical supranuclear saccade (VSSP), gaze palsy (VSGP), and epilepsy [17]. VSSP and VSGP are commonly present in infantile cases and serve as a diagnostic clue [80,81,82]. Children with mutations in the *NPC2* gene often develop respiratory insufficiency due to pulmonary alveolar lipoproteins.

*Therapy:* A holistic and multidisciplinary approach, combining disease-modifying, symptomatic treatments, and non-pharmacological approaches needs to be offered, with the aim to counteract the ongoing neurodegeneration/neuroinflammation and to improve the patients’ quality of life. Miglustat (OGT 918, *N*-butyl-deoxynojirimycin) is the only disease-modifying treatment for NPC [16], approved by the EMA, Health Canada, and regulatory authorities of other countries (Japan) but not the FDA. It is an iminosugar that crosses the blood brain barrier (BBB) and leads primarily to a substrate reduction. However, miglustat may not be used for patients who lack neurological symptoms or for those with an advanced disease state. Miglustat inhibits glucosylceramide synthase, which is needed in the initial stages of glycosphingolipid synthesis, also acting as a non-lysosomal glucosylcerebrosidase (Gba2) inhibitor. The initial clinical trial for NPC was positive [16], followed by several positive longitudinal cohort studies over 2–8 years assessing the neurological state of patients with NPC [83,84,85,86]. The less severely affected juvenile- and adult-onset patients tend to benefit more than the infantile-onset population. The VSGP/VSSP [80,81] also improves or at least stabilizes under treatment [83], as well as dysphagia [87,88].

One of the newly developed, but not yet approved drugs for NPC is *N*-acetyl-*L*-leucine (ALL) [89]. It crosses the BBB. It has beneficial effects not only on motor but also on disease-specific signs and symptoms and on the quality of life of patients with NPC without serious adverse events [18,19,90]. Studies in the *Npc^1−/−/^* mouse model have shown both symptomatic and disease-modifying effects of ALL [42]. A multinational, Phase III, double-blind, randomized, placebo-controlled, crossover trial with ALL for NPC has been completed but the results have not been made public, yet (NCT05163288) [91].

Other substances are in development for NPC, such as chaperone-treatment with arimoclomol [92], AZ-3102 [79], efavirenz [93] or cyclodextrin [94], and are at different development stages. A phase 2, 12-week study evaluating the safety, tolerability, pharmacokinetics, and pharmacodynamics of oral AZ-3102 in patients with NPC or GM2 gangliosidosis (see above) has just been initiated (NCT05758922). The aim of this short study is to determine the clearance of AZ-3102 from the body and to identify the target dose for planned phase 3 studies.

### 3.7. Sialidosis (ORPHA:309294)

Sialidosis or neuraminidase deficiency (mucolipidosis I) is an autosomal recessive disorder caused by a defect in the enzyme sialidase (neuraminidase). Two main clinical variants have been described that may be accompanied by cerebellar ataxia. Sialidosis type I begins in adulthood, the primary features being myoclonic epilepsy and bilateral macular cherry-red spots. Sialidosis type II appears in childhood, comprising skeletal dysplasia, intellectual disability, and hepatosplenomegaly.

*Therapy:* There is no ERT for this condition; the treatment is limited to supportive care and symptomatic relief. Myoclonic epilepsy is often pharmacoresistant, partially responding/not responding to a ketogenic diet, further complicating the management of this disease with worsening cognition [95]. Recently, perampanel has been suggested as efficacious for treating myoclonic epilepsy in this condition [96], as it is in other progressive myoclonic epilepsies, such as Lafora disease [97], Unverricht-Lundborg disease [98], and dentatorubral-pallidoluysian atrophy [99]. Perampanel is a non-competitive, selective α-amino-3-hydroxy-5-methyl-4-isoxazolepropionic acid (AMPA) glutamate receptor antagonist. AMPA receptors may play a pivotal role in the pathophysiology of epilepsy. Symptomatic treatment of cerebellar ataxia, if present, is indicated.

## 4. Disorders of Carbohydrate Metabolism

### 4.1. Galactosemia (ORPHA:79239 and OMIM: 230400)

Classic galactosemia (CG) is caused by mutations in the *GALT* gene, leading to a profound deficiency of galactose-1-phosphate uridylyltransferase (GALT), the central enzyme of galactose metabolism. GALT deficiency leads to an inability to metabolize galactose. Due to the high galactose content in mammalian milk, newborns with CG present with acute intoxication syndrome, a life-threatening condition. Thus, CG is included in the newborn screening programs of several countries. For the diagnosis, GALT enzyme activity is measured in red blood cells (absent or significantly decreased), and/or *GALT* gene analysis is performed [100]. Individuals with CG develop cognitive impairments and motor difficulties in 18–45% of cases [100,101,102,103]. They range from mild positional tremors to pronounced cerebellar ataxia, myoclonus, dystonia, and ocular motor abnormalities [104,105,106]. Neurological symptoms are more frequently reported in adult age; however, the incidence of subtle neurological signs in pediatric patients may have been underestimated in the past.

*Therapy:* A life-long galactose-restricted diet eliminating sources of lactose and galactose from dairy products but permitting galactose from non-milk sources that contribute minimal dietary galactose is required [100]. All types of fruits and vegetables, unfermented soy-based products, mature cheeses (with galactose content < 25 mg/100 g), and the food additives sodium or calcium caseinate are permitted in the CG-specific diet. Although higher in galactose, all fermented soy-based products can be allowed in small quantities as part of the diet [100]. Speech therapy, cognitive training, and treatment of the hypogonadotropic hypogonadism and primary ovarian insufficiency should be initiated if needed. Motor functions should be assessed and physical therapy should be considered if needed. However, CG-specific guidelines for the treatment of motor abnormalities are still lacking. Trihexylphenidyl has been used to treat tremors [104].

### 4.2. GLUT1 Deficiency (ORPHA:71277 and OMIM: 606777)

Impaired facilitated diffusion of glucose across the BBB by the glucose transporter family member 1 (GLUT1) leads to the autosomal dominant Glut1 deficiency syndrome with acute and/or chronic brain energy failure. Depending on the severity of the defect, clinical manifestations range from early-onset epileptic encephalopathy and psychomotor retardation with microcephaly to milder forms with fluctuating movement disorders (including ataxia, tremor, dystonia, chorea and/or spasticity, in variable combinations, or late-onset absence epilepsy). Abnormal eye–head movements in the context of primarily unspecific epilepsy and psychomotor retardation should prompt measurement of the glucose liquor-to-plasma ratio to ascertain hypoglychorrhachia. The diagnosis is confirmed by genetic analysis of the *SLC2A1* gene.

*Therapy*: The treatment of choice is a ketogenic diet. Its rational is to circumvent the cerebral glucopenia by providing ketone bodies as an alternative source of energy to the brain. It should be started as early as possible as it can prevent irreversible damage [107].

### 4.3. Cerebral Creatine Deficiency (ORPHA:79172)

Cerebral creatine deficiency (CCD) is a group of rare diseases caused by mutations in the *AGAT* (CCD syndrome 3, CCDS3), *GAMT *(CCDS2) or X-linked *SLC6A8* (CCDS1) genes, leading to a disturbed synthesis or transport of creatine within the cell. The creatine metabolism plays an essential role in energy turnover, thus CCD leads to early alterations of the mitochondrial proteomic landscape [108]. During periods of high energetic demand, creatine kinases catalyze the transfer of the high-energy phosphate group in phosphocreatine to ADP, allowing for the rapid generation/regeneration of ATP, thereby maintaining the energetic supply required for cellular function [109]. The half of the creatine not taken in through diet is endogenously synthesized by the conversion of glycine and arginine into guanidinoacetate (and ornithine) through the enzyme arginine:glycine amidinotransferase (AGAT) and the subsequent transformation of guanidinoacetate into creatine through the enzyme guanidinoacetate methyltransferase (GAMT). Creatine can then be transferred into brain and muscle tissue using the creatine transporter SLC6A8 [110], where it is converted to phosphocreatine by the creatine kinase acting as a cytoplasmic ATP buffer. The brain is the most affected organ in CDS because it has its own creatine production with high levels of AGAT and GAMT expression, particularly in the cortex [111]. The common clinical presentation in CCDS includes intellectual disability, expressive speech and language delay, autistic-like behavior, and epilepsy. However, each CCD has a unique phenotype. Ataxia is mainly found in GAMT deficiency.

*Therapy:* AGAT and GAMT deficiencies are treatable but not curable conditions. Long-term creatine supplementation usually restores brain creatine levels and improves clinical features but the efficacy of this treatment is limited by the low permeability of the BBB for creatine. Successful treatment strategies (e.g., chaperones or cyclocreatine [112,113]) for creatine transporter deficiency are much needed. The current treatment strategy is simply the supplementation of creatine, arginine, and/or glycine, with limited success. In GAMT deficiency, the treatment consists of a low-protein and arginine-restricted diet as well as ornithine supplementation aimed at reducing levels of neurotoxic guanidinoacetate, along with creatine supply [114]. Safety monitoring and the evaluation of treatment effects is required in all patients. CCDSs may be responsible for a considerable fraction of the diagnoses’ ‘mental retardation of unknown etiology’ in children and adults. Guanidinoacetic acid (GAA), a direct metabolic precursor of creatine, has recently been suggested as a possible alternative to creatine to increase brain creatine levels in experimental medicine. AGAT patients might benefit from oral GAA due to upgraded bioavailability and convenient utilization of the compound. However, possible drawbacks (e.g., brain methylation issues, neurotoxicity, and hyperhomocysteinemia) should be carefully considered as well [115].

## 5. Congenital Disorders of Glycosylation

A number of genetic glycosylation defects (CDGs) can lead to cerebellar dysfunction and ataxia. Phosphomannomutase 2 deficiency is the most common example.

### Phosphomannomutase 2 Deficiency (ORPHA:79318 and OMIM: 212065)

Phosphomannomutase 2 deficiency (PMM2-CDG or CDG1a) is caused by a mutation in the *PMM2* gene, leading to impaired protein *N*-glycosylation. It is the most common congenital disorder of glycosylation. PMM2 catalyzes the conversion of mannose-6-phosphate to mannose-1-phosphate, which is a precursor of guanosine diphosphate mannose (GDP-Man) and dolichol-P-mannose (Dol-Man), which are the donors of mannose in the endoplasmic reticulum. PMM2 deficiency leads to hypoglycosylation of various glycoproteins. The disease is multisystemic and heterogeneous [116]. It is usually diagnosed in childhood as it causes developmental delay. In patients up to 10 years of age, stroke-like episodes can occur. In adulthood, the leading symptoms include cerebellar ataxia and learning difficulties [117]. Neuro-ophthalmological and -otological manifestations include impaired smooth pursuit, nystagmus, ocular flutter, ocular motor apraxia, impaired optokinetic nystagmus, and impaired vestibulo-ocular reflexes [118,119,120,121]. These symptoms point toward the cerebellopontomesencephalic regions being affected. Strabismus and nystagmus might be secondary to visual impairment, even though they can be found in patients with PMM2-CDG with normal vision as well [117,121].

*Therapy:* Currently, there is no approved treatment for PMM2 [116]. A long-term mannose supplementation (≥2 years) led to improved protein glycosylation in the majority of patients, was well-tolerated, and suggested possible clinical improvements, which would need to be investigated with a prospective clinical trial [122]. However, short-term dietary supplementations with mannose at 100 mg/kg body weight (b.w.) every 3 h over lll9 days or 0.17 g/kg b.w. every 3.5 h over a period of 6 months [123,124], as well as a continuous i.v. mannose infusion of 5.7 g/kg b.w. over a period of 3 weeks [125], had failed to show improvements in glycosylation patterns or clinical benefits in previous studies. Acetazolamide, a carbonic anhydrase inhibitor, has been used to treat cerebellar manifestations of PMM2-CDG based on the hypothesis of a gain-of-function effect on the calcium channel CaV2.1 secondary to dysglycosylation, which might be attenuated by this treatment [126]. This approach has had some success, independently of the conjectured mechanism, as acetazolamide appears to improve ataxia due to unrelated diseases, such as CTX (see below) or mucopolysaccharidosis type 3. Currently, investigated agents include the repurposing of epalrestat, an aldose reductase inhibitor, which has been approved for the treatment of diabetes-related neuropathy in some countries [127] (NCT04925960) as well as a mannose-1-phosphate replacement therapy GLM101 (NCT05549219).

## 6. Disorders of Lipid Metabolism

### 6.1. Abetalipoproteinemia (ORPHA:14 and OMIM: 200100)

Abetalipoproteinemia (Bassen-Kornzweig disease) is a neurodegenerative disease caused by a defect in the microsomal triglyceride transfer protein that is critical for the transfer and metabolism of fats in the body. It presents with progressive cerebellar ataxia, seizures, and abnormal eye movements, including ophthalmoplegia with dissociated nystagmus, saccadic slowing, and hypometria [1,128]. Systemic symptoms, including symptoms of fat malabsorption steatosis and abnormal liver transaminases, also occur. Abetalipoproteinemia often presents in the first or second decade of life but may also start in adulthood. The biochemical hallmarks of this disease are very low cholesterol and triglycerides as well as absent ApoB. Low vitamin E leads to progressive neurological symptoms. The diagnosis is confirmed by *MTTP* gene analysis.

*Therapy*: Early intervention with a low-fat diet combined with reduced long-chain fatty acids and fat-soluble vitamin supplements (including 100–150 mg/kg/d of α-tocopherol and vitamin A) can prevent the onset of symptoms or ameliorate the disease [6].

### 6.2. Hypobetalipoproteinemia (ORPHA:31154)

Hypobetalipoproteinemias (HBL) describes a group of diseases characterized by reduced plasma concentrations of low-density lipoprotein-cholesterol (LDL-C) and by intestinal lipid malabsorption and fat-soluble vitamin deficiency. The most common primary monogenic HBL is familial HBL (FHBL), which is due to a defect of the *APOB* gene and has an autosomal co-dominant mode of inheritance [129]. Heterozygous FHBL typically has a rather mild clinical presentation, sometimes including a fatty liver [130]. In contrast, homozygous HBL can have very severe neurological consequences if left untreated, including severe ataxia, myopathy, dysarthria, absent reflexes, retinal degeneration, neuropathy, coagulopathy, hepatic steatosis, progressive demyelination of the central nervous system, lack of responsiveness to local anesthesia, psychomotor retardation, and more, leading to various impairments and a shorter lifespan. An individual genetic diagnosis can facilitate prognosis and treatment planning [131].

*Therapy:* A low-fat diet and vitamin E supplementation in moderate form. However, these measures cannot control or cure HBL, which is why regular follow-ups evaluating the symptoms and compliance with the diet are necessary [132].

### 6.3. Cerebrotendinous Xanthomatosis (ORPHA:909 and OMIM: 213700)

Cerebrotendinous xanthomatosis (CTX) is caused by a deficiency of mitochondrial sterol 27-hydroxylase (CYP27) leading to the accumulation of cholestanol (toxic for the CNS) and cholesterol in tendons, the CNS, and the lenses [133]. Symptoms may develop at any stage of life. The disorder is characterized by solid deposits known as xanthomas near large tendons, cerebellar ataxia, pyramidal, and extrapyramidal signs, which manifest in the second or third decade of life [134,135]. Based on a large-exome human analysis, this disease appears to be underdiagnosed [136]. The typical symptoms are progressive ataxia with spasticity. Other symptoms may include neuropathy, cognitive decline, seizures, and cataracts (which often appear early and may even be the first sign). In addition, ocular motor deficits can be present, including abnormal pursuit, increased saccadic intrusions, multistep saccades, and antisaccades [137].

*Therapy:* A life-long replacement therapy with the bile acid chenodeoxycholic acid should be started as soon as possible, ideally immediately following a positive newborn screening test, in order to prevent or reverse neurological symptoms [133]. A combination of this first-line therapy with an HMG-CoA reductase inhibitor (statin) is effective [138,139]. Chenodeoxycholic acid works by binding to and reducing the accumulation of cholestanol and related metabolites that are damaging to the CNS [134]. The starting dosage in adults is 250 mg tid, which can subsequently be increased to 1000 mg/day if the serum cholestanol and/or urine bile alcohols remain elevated. In the pediatric population, the starting dose is 5 mg/kg/day tid. In neonates, the safety and efficacy of the drug have not yet been established [140].

## 7. Disorders of Mineral, Metal, and Vitamin Metabolism

### 7.1. Ataxia with Vitamin E Deficiency (ORPHA:96 and OMIM: 277460)

Ataxia, due to vitamin E deficiency, can have various causes, such as mutations in the *TTPA* gene, which is responsible for the alpha-tocopherol transfer protein. Several hereditary metabolic ataxias are associated with vitamin E deficiency. However, secondary vitamin E deficiency from a variety of other, non-hereditary causes (e.g., malabsorption and malnutrition) can lead to ataxia as well [141,142]. This ataxia is slowly progressive, accompanied by neuropathy, and may resemble Friedreich ataxia. Some patients present with retinitis pigmentosa [143] or dystonia [144]. Horizontal microsaccadic oscillations, interfering with smooth pursuit, saccades, and optokinetic nystagmus as well as hyperventilation-induced ocular flutter and hippus have been described as well [145].

*Therapy:* Lifelong and early-initiated oral supplementation of high-dose vitamin E in a dosage of 800–1500 mg a day aims at normal plasma levels of vitamin E, which leads to neurological improvements [146]. Nonetheless, recovery may be slow and incomplete [147]. Heterozygotes are clinically healthy but have serum vitamin E concentrations 25% lower than normal.

### 7.2. Multiple Carboxylase Deficiency (ORPHA:148)

Multiple carboxylase deficiency (MCD) can either be due to biotinidase or holocarboxylase synthetase deficiencies. Biotinidase deficiency is caused by mutations in the *BTD* gene. Biotinidase catalyzes the removal of protein-bound biotin to generate free biotin. Its deficiency leads to decreased recycling and eventually to deficiency of biotin, an important co-factor of a number enzymes, especially carboxylases which are involved in gluconeogenesis, the catabolism of several branch-chain amino acids and fatty acid synthesis [148,149]. Manifestations can occur insidiously or acutely depending on the residual biotinidase activity (<10% is considered complete deficiency), mostly during childhood (juvenile-onset form) [14,150]. Symptoms include cerebellar ataxia, ketoacidosis, dermatitis, hearing loss, optic atrophy, seizures, myoclonus, nystagmus, and intellectual disability [151]. In some countries, biotinidase deficiency is part of the newborn screening program. Mutations in the holocarboxylase synthetase gene generally cause the severe infantile-onset form of the disease. This enzyme catalyzes the fixation of biotin to inactive apocarboxylases, producing four active carboxylases (including pyruvate carboxylase).

*Therapy:* Biotin 5–20 (−40) mg/day for prevention or improvement in signs and symptoms. Individuals with biotinidase deficiency, who are screened and treated early and continuously, usually remain completely asymptomatic.

### 7.3. Wilson Disease (ORPHA:905, OMIM: 277900)

Wilson’s disease is caused by mutations in the *ATP7B* gene, leading to a disturbance of a copper transporter and thus to pathological storage of copper in the liver and brain [152]. The first symptoms appear in childhood or young adulthood, including dystonia, parkinsonism, and/or tremor, but predominant ataxia or spasticity cases have been described.

*Therapy:* Chelation treatment with penicillamine or trientine needs to be initiated as soon as possible to prevent (further) neuronal damage. A reduction in intestinal copper uptake can be achieved with zinc administration. In addition to chelation treatment, pyridoxine supplementation is necessary. In some cases, a liver transplantation might become necessary. Dystonia/parkinsonism due to a manganese transporter defect is a more recently described disorder which is treated using a similar strategy that involves chelation of heavy metals with ethylene diamine tetra-acetic acid (EDTA) [153].

## 8. Disorders of Mitochondrial Energy Metabolism

### 8.1. Pyruvate Dehydrogenase (PDH) Deficiency (ORPHA:79243 and OMIM: 312170)

The pyruvate dehydrogenase complex (PDH) is composed of several subunits. Its deficiency is caused by a number of pathogenic variants in one of its genes, most commonly the *PDHA1* gene. It is characterized by lactic acidosis, seizures, impaired cognition, intermittent cerebellar ataxia, and spasticity [154] due to reduced energy production.

*Therapy:* A ketogenic diet, defined by high fat but very low carbohydrate consumption, leads to increased fatty acid oxidation and increased production of ketone bodies [155,156], providing an alternative energy-generating pathway that bypasses the dependency on glycolysis. The disease signs and symptoms can partially be alleviated with a ketogenic regimen, though not completely resolved [157]. A trial with high dose thiamine should be performed in all patients, but generally, only mild and late-onset PDH deficiencies will be responsive. An alternative treatment option is dichloroacetate, which reduces signal transduction by the hypoxia-inducible factors and is an inhibitor of E1 kinase, thereby stimulating (residual) PDH complex activity. It has been used in cancer cells [158]. It might have an effect on most *PDHA1* pathogenic variants and can act as a temporary treatment to reduce lactic acidosis, a common symptom of PDH deficiency [159]. Phenylbutyrate has been used in MSUD, where it acts on the kinase of the BCKAD, a member of the same family of PDH kinase [29,160]. It prevents the PDH kinase from phosphorylating the PDH complex, allowing the complex to remain active. However, it can only be used in patients with certain missense variants (p.P221L, p.R234G, p.G249R, p.R349C, and p.R349H) of the PDHA1 protein. In addition, carnitine or lipoic acid supplementation can be advantageous.

### 8.2. Coenzyme Q 10 (Ubiquinone) Deficiency (ORPHA:139485 and OMIM: 612016)

Coenzyme Q10 (CoQ10) plays an important role in electron transport, especially in oxidative phosphorylation (OXPHOS) in that it distributes electrons between the various dehydrogenases and the cytochrome segments of the respiratory chain, central to ATP production. Its deficiency is an important cause of treatable chronic progressive ataxia. Primary CoQ10 deficiency, due to a genetic defect in coenzyme Q biosynthesis, is a clinically heterogeneous condition. Several disease genes have been identified. The *CABC1* gene, also called *ADCK3*, is one of the numerous genes involved in the ubiquinone biosynthesis pathway. Patients with a mutation in the CABC1 gene feature a progressive neurological disorder with cerebellar atrophy and seizures [161]. A muscle biopsy typically shows decreased levels of ubiquinone. A similar condition is *COQ8A*-ataxia but it is a mitochondrial disease in which defect coenzyme Q10 synthesis leads to dysfunction of the respiratory chain. The disease usually presents in childhood with progressive ataxia accompanied by developmental regression and cerebellar atrophy. Given the heterogeneous phenotype, it may be hard to distinguish from other mitochondrial diseases or a wide spectrum of childhood-onset cerebellar ataxias. Diagnostic confirmation is performed by genetic analysis, usually whole exome sequencing (WES) owing to the many differential diagnoses and the difficult biochemical testing of these disorders.

*Therapy:* The rapid diagnosis of this potentially treatable group of conditions is paramount. Treatment entails high dose CoQ10 substitution for life, albeit with variable success. Reported doses range from 60 to 700 mg/day (Class IV) [161,162,163]. *COQ8A*-ataxia is a potentially treatable condition with supplementation of the coenzyme Q10 as the main therapy. However, 50% of patients may not respond to the treatment.

### 8.3. Mitochondrial and Nuclear Defects of the Respiratory Chain Machinery

Many syndromal mitochondrial diseases may present with ataxia as an important symptom. Examples include myoclonic epilepsy with ragged red fibers (MERRF), mitochondrial encephalopathy, lactic acidosis, stroke-like episodes (MELAS), neuropathy, ataxia, retinitis pigmentosa (NARP), maternally inherited Leigh syndrome (MILS), and Kearns–Sayre syndrome. In addition, ataxia may also be associated with defects in one of the genes coding for a factor of the mitochondrial maintenance machinery, including tRNAs and their synthetases and polymerases and proteins directly involved in the mitochondrial energy metabolism. The diagnosis can be suspected from the clinical combination of symptoms with indicative biochemical markers, such as increased lactate (though present in less than 50% of cases), pyruvate, and Krebs cycle metabolites present in urine. The diagnosis can further be substantiated by a functional and genetic (mtDNA) workup in fibroblasts or preferably in cells from affected and generally energy intensive tissues such as muscle or kidney (alternatively urothelial cells). Genetic analysis of the more than 1500 nuclear genes with mitochondrial functions and the mitochondrial genome establishes the diagnosis [164].

*Therapy*: Despite many years of research investigating numerous different compounds, no specific treatment for “classical” mitochondrial diseases is available today or on the horizon [165]. In clinical practice, most often a “cocktail” of vitamins is prescribed, aiming at increased residual respiratory chain function or decreased accumulation of reactive oxygen species. However, this approach has uncertain effects, apart from some specific defects of co-factors, such as coenzyme Q10 or riboflavin-related disorders, with a primary or secondary deficiency of the substituted factor [12,13].

## 9. Peroxisomal Disorders

### 9.1. Zellweger Spectrum Disorders

The peroxisome biogenesis disorders in the Zellweger spectrum disorders (PBD-ZSD) range from severe (Zellweger Syndrome, ZS) and intermediate (Neonatal Adrenoleukodystrophy, NALD) to mild (Infantile Refsum disease, IRD) phenotypes. They are characterized by a defect in peroxisome formation and caused by mutations in one of 13 *PEX* genes [166,167]. As a result of the defect in peroxisome formation, both catabolic and anabolic pathways are involved. Despite the heterogeneity, the metabolic profile involves very long chain fatty acids (VLCFAs), phytanic- and pristanic acid, C27-bile acid intermediates and pipecolic acid in plasma, and a deficiency of plasmalogens in erythrocytes [168]. Based on when the disease starts, early-infantile, late-infantile/childhood-onset, and juvenile/adult onset versions are distinguished. The later the disease manifests, the milder the phenotype is.

*Therapy:* No effective causal treatment is known to date. A number of small molecules, including betaine (an organic chemical compound with both a positive and a negative charge in its molecule but uncharged on the surface), seemed promising [169,170,171,172]. However, the details of the conducted clinical trial (NCT01838941) have not been reported elsewhere suggesting negative results. The long-term administration of cholic acid (ISRCTN96480891) in a large cohort of ZSD patients did not significantly improve clinically relevant parameters despite reducing the toxic C_27_-bile acid intermediates [173] and beneficial effects in smaller cohorts [174,175]. Considering the gene therapy, this must be investigated in the *PEX1* mouse model [176] before a human trial can be initiated.

### 9.2. Refsum Disease (ORPHA:773, OMIM: 266500)

Refsum disease is caused by a deficiency of phytanoyl-CoA hydroxylase (PhyH) due to mutations in the *PHYH* gene, i.e., an inability to degrade the branched-chain fatty acid, phytanic acid. The main neurological feature is progressive cerebellar ataxia, presenting in late childhood or early adulthood with the additional triad of ichthyosis, retinitis pigmentosa, and neuropathy.

*Therapy:* A diet of reduced phytanic acids can prevent or reverse symptoms of Refsum disease [62,63].

### 9.3. X-Linked Adrenoleukodystrophie (ALD) (ORPHA:43 and OMIM: 300100)

ALD is a peroxisomal disorder of beta-oxidation. It includes a spectrum of phenotypes, such as cerebral ALD (in males of 3 years and older), isolated or combined adrenal insufficiency (in males aged 6 months or older), and also the less severe adult form adrenomyeloneuropathy (AMN; in males usually after 18 years and females after 40 years of age) [177]. The age of onset and severity of clinical presentation vary; the phenotype does not correlate with the type of mutation and within the same family, different phenotypes can occur [178]. Once affected boys have reached 10 years of age, about one-third of them will have developed brain disease and half of them adrenal disease. The lifetime risk for adrenal insufficiency is 80% [179,180]. Most affected males will develop some neurological symptoms. ALD/AMN is an X-linked disorder caused by mutations in the adenosine triphosphate (ATP)-binding cassette (ABC), subfamily D, member 1 gene (*ABCD1*), which encodes an ABC transporter [181]. ALD leads to an accumulation of very long-chain fatty acids (VLCFAs) in all tissues, given the ABC transporter helps form the channel through which VLCFAs normally move into the peroxisome. ALD protein distribution correlates with regions of high metabolic activity (heart, skeletal muscle, and liver) and with critical neural regions [182]. The cerebral forms of ALD are characterized by diverse immune responses. The profound mononuclear response is characterized by microglial activation followed by apoptosis and differs from that in multiple sclerosis [183]. The inflammatory demyelination usually first affects the splenium of the corpus callosum and the occipitoparietal regions with asymmetric progressions toward the frontal or temporal lobes. Sometimes, lesions may be seen in the brainstem as well, especially with the involvement of the pons. Usually, the spinal cord remains unaffected except for bilateral corticospinal tract degeneration [184]. The primary manifestation of AMN Is spinal cord dysfunction with progressive stiffness and weakness of the legs (spastic paraparesis), abnormal sphincter control, neurogenic bladder, and sexual dysfunction. Motor abnormalities are often preceded by gonadal dysfunction. AMN is also a progressive cerebellar disorder. In males with AMN, numbness and pain from polyneuropathy may occur. The majority of childhood ALD patients and adult AMN patients have adrenal insufficiency. Cerebral symptoms such as cognitive decline, behavioral abnormalities, visual loss, impaired auditory discrimination, feeding difficulties, poor coordination, ataxia and seizures often develop over the course of the disease. Primary adrenal insufficiency may be the initial manifestation of ALD and can remain its only sign. It may present as fatigue, nonspecific gastrointestinal symptoms, vomiting, weakness, morning headaches, or, in some cases, as an acute Addisonian crisis [185]. Hyperpigmented skin due to increased ACTH secretion is possible. Fasting hypoglycemia may occur [186]. The first step in diagnosing ALD/AMN is the VLCFA panel, which is more sensitive in male patients than in female carriers [187]. Typically, these three VLCFA parameters are included: C22:0, C24:0, and C26:0 levels, as well as the ratios of C26:0 to docosanoic acid (C26:0/C22:0) and C24:0/C22:0 [188]. Genetic testing confirms the diagnosis. In addition, adrenal function is evaluated by morning cortisol and ACTH or stimulation testing. Confirmed ALD patients without neurological symptoms are advised to undergo regular surveillance neuroimaging. Prenatal DNA-based testing is available [189].

*Therapy:* Treatments differ as they are targeted to specific phenotypes. First line treatment for boys with early stages of cerebral ALD is hematopoietic stem cell transplant (HSCT) [190]. However, HSCT should not be undertaken in advanced neurologic disease or presymptomatic patients. Gluco- and mineralocorticoid replacement can treat adrenal insufficiency, which remains untreated by HSCT [191]. Treatment for adults with AMN is primarily supportive and similar to other myelopathies. HSCT does not seem beneficial to AMN patients without cerebral involvement as it may even exacerbate myelopathy symptoms [192]. The aim is the prevention and treatment of complications of spinal cord disease. Advanced ALD and AMN are treated supportively [193].

## 10. Disorders of Purine or Pyrimidine Metabolism

### Lesch-Nyhan Disease and Its Variants (ORPHA:510 and OMIM:300322)

Lesch-Nyhan disease is an X-linked inherited disease predominantly affecting men and very rarely women [194]. It is caused by mutations in the *HGPRT1 gene* [195]. The risk to an affected child’s siblings depends on the carrier status of the mother, not the father. Carrier women have a 50% chance of transmitting the *HPRT1* variant in each pregnancy. Sons will inherit the disease and daughters will be carriers [196].

The disorder arises from a deficiency of the purine salvage enzyme hypoxanthine–guanine phosphoribosyltransferase (HGprt). The disease is characterized by elevated uric acid, increasing the risk for nephrolithiasis, renal insufficiency up to renal failure, crystalluria, or gout with tophi [195]. It can be efficiently managed with allopurinol treatment but this treatment has no impact on the neurocognitive and behavioral consequences of the disease. The phenotype is variable and depends on the level of residual HGprt activity. The classical phenotype encompasses low or undetectable levels of the HGprt enzyme, rarely with enzyme activity above 1%. These patients feature the classic phenotype with hyperuricemia, dystonia, dysarthria, ocular motor involvement (including inability to initiate voluntary saccades (ocular motor apraxia) and impaired smooth pursuit [197]), intellectual disability, and self-injurious behavior leading to self-mutilation. Patients with HGprt enzyme activity that ranges from 1.6 to 8% of the normal level present with hyperuricemia without self-injury. Enzyme activity levels that exceed 8–10% of the norm typically present with hyperuricemia without self-injury. They may display ataxia and mild cognitive impairment depending on the enzyme levels [195].

*Therapy:* Multimodal treatment is necessary, including pharmacotherapy, physical therapy, behavioral therapy, and psychiatric interventions. Protective equipment and dental management counteract self-mutilation. Pharmacological management focuses on mood and anxiety stabilization, including atypical antipsychotics (e.g., risperidone [198], antidepressants, carbamazepine [199], or gabapentin [200]). Given the involvement of the basal ganglia and the associated L-Dopa deficiency, L-Dopa was administered. However, instead of the anticipated amelioration, this lead to an exacerbation of the motor symptoms [201]. S-adenosylmethionine (SAMe) has shown beneficial effects in patients who can tolerate the drug [202]. Deep brain stimulation was shown (in several case reports and case series) to reduce or eliminate self-injury and aggression and to modify dystonia in some cases [203,204,205,206].

## 11. Urea Cycle Enzyme Deficiencies

The metabolic pathway that allows ammonia detoxification (by conversion from peripheral (muscle) and enteral sources (protein ingestion) into urea) for excretion from the body is called the urea cycle. Urea cycle disorders (UCDs) are deficiencies of an enzyme (or transporter) in that pathway [207], namely the following:Carbamyl phosphate synthetase I (CPSI) deficiency;Ornithine transcarbamylase (OTC) deficiency;Argininosuccinate synthetase (ASS) deficiency (also known as classic citrullinemia or type I citrullinemia (CTLN1);Argininosuccinate lyase (ASL) deficiency (also known as argininosuccinic aciduria);*N*-acetyl glutamate synthetase (NAGS) deficiency;Arginase deficiency.

The severe neonatal forms of all UCDs, except for arginase deficiency, lead to hyperammonemia and potentially lethal metabolic decompensation in the days following birth. Survivors frequently suffer from severe neurological injuries. UCDs are autosomal recessively inherited, except for the x-linked OTC deficiency. In affected females, the severity of clinical symptoms depends on the pattern of X-inactivation in the liver (lyonization) and can be almost as severe as in males. The majority of patients become symptomatic in early childhood. However, a disease onset in later childhood or in adulthood is possible. Common symptoms of UCD are frequent vomiting, poor appetite, protein aversion, and food refusal [208]. Other symptoms are neurological (decreased level of consciousness, altered mental status, abnormal motor function, and seizures) and gastrointestinal (vomiting, poor feeding, diarrhea, nausea, and constipation). Severe deficiencies manifest 24–48 h postnatally in newborns, who seemed initially well. Due to the protein in human milk, the newborn becomes symptomatic soon after birth, with a clinical presentation similar to sepsis [209]. The accumulation of ammonia, glutamine, and other metabolites leads to cerebral edema [210]. Patients have a lifelong risk of increased catabolism leading to metabolic decompensation. Intellectual disability and developmental delays are typical [211]. Atypical symptoms are most commonly seen with partial enzyme deficiencies (frequently with OTC deficiency), including sleep disorders, psychiatric illness, headaches, anorexia, vomiting, lethargy, ataxia, and behavioral abnormalities after increased protein intake or elevated stress levels [20,212]. These patients often prefer vegetarian diets to reduce dietary protein intake. Elevated plasma ammonia concentration (>100 to 150 µmol/L) is the laboratory hallmark of UCDs. The plasma ammonia concentration needs to be measured in an arterial or venous blood sample; capillary blood is not reliable. Mild elevations of the ammonia concentration threshold are to be interpreted based on the clinical presentation and need to be double-checked with a follow-up measurement. Elevated plasma ammonia concentration with normal blood glucose and anion gap suggests a high probability of a UCD. In patients with partial defects, the ammonia needs to be measured at the time of decompensation as it may be within the normal range during the well-compensated periods [213]. Equally, plasma amino acids as well as urinary organic and orotic acids need to be analyzed during symptomatic episodes. Neuroimaging findings tend to vary with the severity and duration of the hyperammonemia [214]. Enzyme analysis and molecular genetic testing can confirm a specific diagnosis (Table 1).

*Therapy:* Therapy can entail a low-protein diet, essential amino acids supplementation, nitrogen scavengers (sodium benzoate or/and phenylbutyrate), carglumic acid in NAGS deficiency, and substitution with L-arginine or L-citrulline in proximal UCD. Emergency management entails high intravenous glucose and nitrogen scavengers. Clinical trials of AVV-mediated gene transfer are currently ongoing (NCT02991144, NCT05345171, and NCT03636438). Another approach is to administer mRNA coding for OTC packaged into lipid-based nanoparticles to adolescent and adult patients with OTCD (phase 1/2 studies: NCT05526066, NCT03767270). Special management of acute and life-threatening hyperammonemia is necessary (Table 1) [7]. Patients with severe disease burden may need a liver transplantation.

## 12. Common Therapeutic Strategies

### 12.1. Physical Therapy

Physical therapy plays a crucial role in managing cerebellar ataxias. A physical therapist can design an individualized exercise program to improve balance, coordination, and muscle strength. There is a robust body of evidence demonstrating the importance of physical therapy and neuro-rehabilitative strategies in cerebellar ataxia [215,216]. The currently ongoing randomized controlled trial “Rehabilitation for ataxia study” [217] (ACTRN12618000908235) investigates the efficacy of a 30-week-long outpatient and home-based rehabilitation program for individuals with hereditary cerebellar ataxia. Physical exercises may include gait training, balance exercises, and stretching to maintain a range of motion. Hippotherapy may also be helpful but reimbursement issues may limit its use in practice. Nevertheless, not all strategies lead to improvements [218]. Patients may benefit from assistive devices depending on the severity of ataxia and its impact on mobility to improve stability and support individuals in maintaining mobility and independence. In case of frequent falls, orthoses and helmets should be considered to prevent fall-related injuries that are a major cause of morbidity and mortality in these patients.

### 12.2. Occupational Therapy

Occupational therapists can help individuals with hereditary cerebellar ataxias develop strategies to adapt to daily living activities. They may provide assistive devices such as braces or splints to support weakened limbs, recommend modifications in the home environment, and suggest adaptive techniques to maintain function and improve independence.

### 12.3. Speech Therapy

Speech and swallowing difficulties are common in hereditary cerebellar ataxias. Speech therapy can help individuals improve their speech clarity, coordination, and swallowing function. Therapists may teach exercises to strengthen the muscles involved in speech and swallowing and provide strategies to compensate for difficulties.

### 12.4. Symptomatic Treatment

As for cerebellar ataxias due to cerebrovascular insult, tumor, or trauma and cerebellar ataxias of other etiologies (including degenerative, autoimmune, etc.), the aim of symptomatic treatment is to improve the load and burden of symptoms (e.g., ambulation, fine motor skills, and speech). The experience of symptoms can differ a lot between patients due to the large variety in symptom etiology and due to individual differences in personality, mental wellbeing, resilience, support systems, education, and means between patients. While medications cannot cure cerebellar ataxias, they may be prescribed to manage specific symptoms, not only for ataxias but also for other signs and symptoms of the aforementioned disease (e.g., dystonia, spasticity, or dyskinesia). Here, we focus on symptomatic treatments of cerebellar ataxia that have already been introduced so far, with varying levels of evidence.

#### 12.4.1. Acetyl-L-Leucine

As mentioned in chapter 2.2 and 2.6, ALL is the L-enantiomer of *N*-Acetyl-*DL*-Leucine (ADLL), an acetylated derivative of the amino acid leucine, which is ubiquitously present in human food [89]. ALL is the active component of the racemate and is currently not being authorized anywhere in the world for the treatment of any condition. ALL and ADLL can cross the blood–brain barrier (BBB). The drug has been shown to be safe and well tolerated [90,219]. ALL’s mode of action is multimodal, leading to increased ATP production, decreasing the superoxide dismutase 1 and mitochondrial reactive oxygen species (ROS), and reducing lipid storage in neuronal and non-neuronal tissues [42]. ALL functions as a pro-drug of leucine, leading to leucine being metabolized to alpha-ketoisocaproate and oxidized in the citric acid cycle. In contrast to leucine, ALL completely lacks toxicity because it is primarily transported by the monocarboxylate transporter (MCT1), not the L-type amino acid transporter (LAT1) [220]. Thus, ALL enables the delivery of more leucine to tissues because the transport via MCT1 is not limited by the competition with the uptake of other essential amino acids, as is the case with the transport via LAT1 [220].

A case series of patients with cerebellar ataxia of various origins treated with ADLL suggested a clinically significant benefit of the treatment [221,222] that has also been supported by neuro-metabolic studies [223,224,225]. However, the evidence for the efficacy of ADLL is somewhat contradictory with a negative placebo-controlled clinical trial that failed to show the improvement in cerebellar ataxia signs and symptoms [226]. This might be due to the heterogeneity of selected cerebellar ataxia subtypes that lead to a higher response variability together with studying a racemate instead of the ALL-specific active compound. Further clinical trials in more homogeneous populations with cerebellar ataxia are needed to evaluate the efficacy of ALL.

#### 12.4.2. Amantadine

Amantadine, a non-competitive *N*-methyl-*D*-aspartate antagonist, which has positive effects on some Parkinsonian features, seems to benefit patients with degenerative ataxias. It has shown improvements in the functional ataxia scoring scale after daily administrations of max. 200 mg/day (Class III and IV) for a few months [227,228] but no significant improvement was found in patients with Friedreich ataxia (Class II) [229].

#### 12.4.3. Aminopyridines and Acetazolamide

Aminopyridines, such as 4-aminopyridine (4-AP), 3,4-diaminopyridine, and prolonged-release 4-aminopyridine (fampridine) are blockers of the so-called Kv1 voltage-activated potassium channels [230], increasing the excitability of neurons, especially of cerebellar Purkinje cells and other cerebellar cells [231,232]. In a calcium channel, “tottering” mouse model of episodic ataxia type 2 (EA2), aminopyridines seem to increase the threshold for attack initiation without mitigating the character of the attack per se [233]. Two randomized, double-blind, placebo-controlled trials have shown significant benefits of treatment, i.e., reduction in attack frequency, and an improved quality of life with aminopyridines in EA2 [234,235]. Acetazolamide, a carbonic anhydrase inhibitor, has been shown to reduce the number of attacks as well (compared to placebo), but it also showed more adverse events compared to fampridine [235]. Unfortunately, fampridine is not yet approved for the indication of EA2 but for gait disturbance in patients with multiple sclerosis [236,237,238]. Importantly, the positive effects of aminopyridines are not restricted to EA2; in a mouse model of spinocerebellar ataxia type 1, they seem to correct early dysfunction and delay the neurodegeneration, thus acting neuroprotectively [239]. There is evidence for improvements in cerebellar ataxic gait under aminopyridines [240] as well as improvements in tremor [241]. Aminopyridines also ameliorate cerebellar downbeat-nystagmus and improve vision in affected individuals [222,242]. The positive effects are seemingly not restricted to the cerebellum: aminopyridines are a promising treatment option for patients with gain-of-function *KCNA2*- and *KCNA1*-encephalopathy [243,244] as well as severe intractable epileptic syndromes with intellectual disability and other neuropsychiatric symptoms.

#### 12.4.4. Omaveloxolone

Omaveloxolone is a potent activator of Nrf2 that has been shown to be effective at improving symptoms and slowing progression of Friedreich ataxia (FA) [245]. The treatment has been approved by the FDA (02/2023) for individuals from 16 to 40 years of age with genetically confirmed FA. FA is not a metabolic ataxia, even though the mutations in the frataxin gene lead to dysfunction in the mitochondria, consequently decreasing the production of adenosine triphosphate (ATP) which leads to a lack of cell energy. An impaired signaling of Nrf2, a transcription factor that regulates the cellular defense against oxidative stress, is one of the key pathophysiological mechanisms. Thus, applying Nrf2 leads to restored function.

The approval of omaveloxolone is supported by the efficacy and safety data from the MOXIe Part 2 trial, a randomized, double-blind, and placebo-controlled study together with the data from the open-label MOXIe extension trial [245,246]. The daily treatment with 150 mg omaveloxolone for 48 weeks was generally well tolerated and resulted in statistically significant improvements in neurological functions. The most common adverse reactions in MOXIe Part 2 (≥20% and greater than placebo) were elevated liver enzymes (AST/ALT), headache, nausea, abdominal pain, fatigue, diarrhea, and musculoskeletal pain.

#### 12.4.5. Riluzole

Riluzole has multimodal effects which include inhibiting excitatory amino acid release and stabilizing the inactivated state of voltage-dependent sodium channels but also opening small-conductance calcium-activated potassium channels (that regulate the firing rate of neurons in deep cerebellar nuclei), thus reducing neuronal hyperexcitability [247,248]. In a double-blind placebo-controlled pilot trial including 40 patients with ataxia of different etiologies, riluzole (100 mg/day) demonstrated a safe risk–benefit profile and a mean decrease in 7 of 100 points on the International Cooperative Ataxia Rating Scale (ICARS) after 8 weeks (Class II) [249]. For reference, the cut-off for a clinically relevant difference is set at five points.

#### 12.4.6. Varenicline

Varenicline is a partial agonist at α4β2 nicotinic acetylcholine receptors. In a randomized controlled trial with SCA3 patients, a dosage of 1 mg twice per day has mainly led to gait improvements (Class II) [250]. However, due to significant drawbacks in the study design, these results need to be interpreted with caution [251].

### 12.5. Genetic Counseling

Hereditary metabolic cerebellar ataxias are caused by genetic, usually autosomally recessive, and rarely X-linked patterns of inheritance, such as in X-linked adrenoleucodystrophy. Genetic counseling may help individuals and families understand the inheritance patterns, the concept of carrying a pathogenic variant, the risk of passing on the condition, and the availability of genetic testing options. This information can be valuable when it comes to family planning for persons of childbearing potential and their families trying to reach a well-informed decision. In addition, non-medical factors, including but not limited to psychosocial factors, personal circumstances, social support, accessibility, financial means, and more need to be considered in the decision-making process as well.

It is highly relevant that individuals affected by a hereditary metabolic ataxia are provided with the opportunity to work closely together with a team of healthcare professionals (including neurologists, metabolic experts, endocrinologists, physical therapists, occupational therapists, psychologists, and other specialists) to tailor the therapy to their individual needs. Additionally, support groups and organizations dedicated to ataxia can provide valuable resources, education, and emotional support for individuals and their families affected by these conditions.

## Figures and Tables

**Table 1 cells-12-02314-t001:** An overview of the diseases discussed in this review. In general, cerebellar ataxia is mainly observable in milder forms of IEM and manifests between late-infantile and adult-onsets. Late-adult onsets in persons above age 60 are rare but possible. Multidisciplinary approaches, combining physical therapy, psychological/behavioral/psychiatric intervention, protective equipment, and pharmacological treatment, incl. disease-modifying (where possible) and symptomatic approaches, should be applied, where necessary. Standardized monitoring for safety and evaluation of treatment effects is required in all patients. Mutation analysis is standard for confirmatory diagnostics.

Disease (in Alphabetical Order), ORPHAcode, OMIM, Prevalence (P)	Causative Gene, Gene Locus, Protein and Mode of Inheritance	Mode of Occurence	Diagnostics	Movement Disorders	Other Clinical Signs	Treatment and Considerations	Outcome
Abetalipoproteinemia (Bassen-Kornzweig), (ORPHA:14, OMIM: 200100) P: 1/1,000,000	*MTTP*, 4q23, Microsomal triglyceride transfer protein (MTTP). Autosomal recessive	Chronic progressive	Plasma Vitamin E	Ataxia, chorea, dystonia, parkinsonism	Retinitis pigmentosa, dementia, seizures, ophthalmoplegia with dissociated nystagmus, saccadic slowing, hypometria, acanthocytosis, fat malabsorption steatosis, abnormal liver transaminases	Dietary fat restriction, reduced long-chain fatty acids, vitamins E (α-tocopherol) 150 mg/kg/day & vitamin A supplements	Early treatment prevents symptoms or ameliorates the disease [6]
Adrenomyeloneuropathy (AMN = late onset form, also in females), (ORPHA:1393, *OMIM:* 300100) P: unknown	*ABCD1*, Xq28, Adrenoleukodystrophy protein-adenosine triphosphate (ATP)-binding cassette (ABC), subfamily D, member 1 gene. X-linked recessive	Chronic progressive	Very long-chain fatty acid (VLCFA): Neuroimaging Adrenocorticotropic hormone dosing or stimulation for adrenal insufficiency Electrophysiology testing for AMN	Spinal cord dysfunction with spastic paraparesis, abnormal sphincter control, progressive incoordination and ataxia	Inflammatory and noninflammatory demyelination, peripheral neuropathy, learning disabilities, behaviour problems, neurologic deterioration, blindness, seizures, adrenal insufficiency, hyperpigmented skin, neurogenic bladder, sexual dysfunction, impaired auditory discrimination, fasting hypoglycemia	Regular follow-up by neuroimaging; at early signs of progression: Hematopoietic stem cell transplantation (HSCT) for (early) cerebral ALD. Primarily supportive for adults with AMN. Gluco- and mineralocorticoid replacement for adrenal insufficiency	Treatment of early cerebral ALD, Considerably higher five-year survival in ALD
Arginase deficiency (Argininemia), (ORPHA:90, *OMIM:* 207800) P: <1/1,000,000	*ARG1*, 6q23, Arginase. Autosomal recessive	Intermittent or chronic progressive	variably n-↑ plasma ammonia Neuroimaging Plasma amino acids (↑↑ Arg, ↑ Cit)/↑ urinary orotic acid analyses Enzyme analyses in erythrocytes	Progressive spastic quatriplegia, rarely ataxia	Chronic vomiting, anorexia, growth failure, developmental delay, somnolence, seizures, sleep disorder, psychiatric illness, headache, lethargy, behaviour abnormalities, central hyperventilation, cerebral edema	Arginine-restricted diet, nitrogen scavengers if needed, essential nutrients, essential amino acids, emergency management. In case of acute hyperammonemia:anticatabolic treatment and nitrogen scavenger.	Depending on the subtype
Argininosuccinic aciduria (ASL deficiency), (ORPHA:23, *OMIM:* 207900) P: 1–9/100,000	*ASL*, 7q11.21, Arginosuccinate lyase, Autosomal recessive	Intermittent or chronic progressive	↑ plasma ammonia concentration ↑↑ Argininosuccinic acide in plasma and urine High plasma citrulline (100–300 µmol/L) ↓ plasma arginine High plasma glutamine Elevated liver enzymes in serum Orotic aciduria Neuroimaging Enzyme analysis in red blood cells or fibroblasts	Ataxia	Chronic vomiting, failure to thrive, developmental delay, somnolence, seizures, sleep disorder, psychiatric illness, headache, lethargy, behaviour abnormalities, central hyperventilation, cerebral edema, Trichorrhexis nodosa, Dry brittle hair	Protein-restricted diet, nitrogen scavengers as needed, arginine-supplementation, essential amino acids and vitamins, emergency management. In case of acute hyperammonemia: anticatabolic treatment and nitrogen scavenger [7].	Outcome depends on the severity of the enzyme defect, the extent and duration of initial and recurrent hyperammonemia, as well as diagnostic delay
Ataxia with CoQ10 (ubiquinone) deficiency (ORPHA:139485, *OMIM:* 612016) See also genetic forms of secondary CoQ10 deficiencies! P: unknown	*ADCK3/SCAR9,* 1q42.13, (potentially other causes of primary CoQ10 deficiency: *PDSS1*, *PDSS2*, *COQ2*, *COQ4*, *COQ5*, *COQ6*, *COQ7*, *COQ9*). All autosomal recessive	Chronic progressive	Decreased coenzyme Q Increased serum and CSF lactate Decreased activity of respiratory complex II + III Decreased activity of respiratory complex I + III Muscle biopsy shows mitochondrial aggregates, lipid droplets, and decreased levels of ubiquinone	Ataxia with exercise intolerance, hypotonia, proximal muscle weakness, Tremor, myoclonic jerks	Developmental regression and cerebellar atrophy, variably seizures, mild mental impairment Lactic acidosis	CoQ10 supplementation, 60 to 700 mg/day reported in the literature, variable response rate	Variable response rate and symptom amelioration
Ataxia with vitamin E deficiency, (ORPHA:96, *OMIM:* 277460) P: 1–9/1,000,000	*TTPA*, 8q13, Alpha-tocopherol transfer protein. Autosomal recessive	Chronic progressive	Plasma vitamin E undetectable High serum cholesterol, triglyceride and beta-lipoprotein Defective liver ‘tocopherol binding protein’ Brain MRI *TTPA* sequencing	Ataxia, rarely dystonia, spasticity	Somatosensory disturbance with loss of proprioception and areflexia	Vitamin E supplements [8,9] Monitor vitamin E levels at regular intervals,	Improvement in symptoms
Cerebral creatine deficiency, (ORPHA:79172) P: <1/1,000,000	*GAMT*, 19p13.3 guanidinoacetate methyltransferase AGAT *SLC6A8*. X-linked recessive but symptoms in females may occur.	Chronic progressive or static/non-progressive	low creatine excretion (but: ↑ creatine/creatinine-ratio in urine of transporter deficient males) ↓ creatine in plasma (not transporter deficiency) Low CSF creatine Low CSF creatinine Deficiency of creatine phosphate in the brain Accumulation of guanidinoacetate in urine, plasma and brain (only GAMT deficiency) Guanidinoacetate methyltransferase (GAMT) deficiency MR-spectroscopy: ↓↓ creatine-peak	Chorea, dystonia, ataxia, hypotonia in infancy	Developmental delay, epilepsy, behavioral abnormalities, myopathy 50% of female carriers with transporter defect have learning and behavioural problems	GAMT: creatine-supplementation, ornithine, +/− Na benzoate. Low-protein diet with restricted arginine AGAT: creatine supplementation, in experimental medicine Guanidinoacetic acid (GAA) intake?; drawbacks e.g., brain methylation issues, neurotoxicity, and hyperhomocysteinemia Creatine transporter deficiency: supplementation of creatine, arginine and glycine (unproven effect). chaperones or cyclocreatine?	Partial improvement in symptoms [10] (the earlier the better)
Cerebrotendinous xanthomatosis, (ORPHA:909, *OMIM:* 213700) P: unknown	*CYP27A1*, 2q33, Mitochondrial sterol 27-hydroxylase. Autosomal recessive	Chronic progressive	Normal to slightly elevated plasma cholesterol Elevated plasma cholestanol Elevated urinary 7 alpha-hydroxylated bile alcohols Sterol 27-hydroxylase deficiency	Ataxia, spasticity, pyramidal and extrapyramidal signs	Xanthomas near large tendons, neuropathy, cognitive decline, seizures, cataracts, ocular motor deficits: abnormal pursuit, increased saccadic intrusions, multistep saccades, antisaccade deficits	Prompt installation of treatment with chenodeoxycholic acid, 250 mg tid, or in patients < 18 years, 5 mg/kg/body weight tid possibly in combination with HMG-CoA reductase inhibitor (statin)	Prevent or reverse neurological symptoms
Ceroid lipofuscinosis, (ORPHA:216) P: 1-9/100,000	Several variants, multiple gene products: *PPT1,* 1p34.2, CLN1: Palmitoyl protein thioesterease-1 (PPT1) deficiency; *TPP1*, 11p15.4, CLN2: tripeptidyl peptidase-1 (TPP1) deficiency, *CLN5*, 13q22.3, CLN5: CLN5 protein deficiency, *CTSD*, 11p15, CLN10: cathepsin D deficiency, *CTSF*, 11q13.2, CLN13: cathepsin F deficiency. All other NCL forms are caused by defects in lysosomal, endoplasmic reticulum or cytoplasmic vesicular transmembrane proteins. Autosomal recessive but CLN4 (autosomal dominant).	Chronic progressive	Clinical clusters of signs and symptoms Histology MR Imaging may be suggestive Genetic testing	Ataxia, myoclonus, spasticity, spastic quadriparesis, parkinsonism	Vision loss, dementia, epilepsy, decreased level of consciousness, depression, speech delay, stuttering dysarthria, developmental plateau, polymorphic seizures	Supportive care paired with pharmacological strategies: Epilepsy: valproate lamotrigine, topiramate, levetiracetam Spasticity: Baclofen CLN2: intrathecal ERT (Recombinant human cerliponase alfa administered into the cerebrospinal fluid via the lateral ventricles)	Supportive, only CLN2 disease is presently treatable
Citrullinemia Type 1 (ORPHA:2475, *OMIM:* 215700) P: 1-9/100,000	*ASS1*, 9q34, Arginosuccinate synthetase. Autosomal recessive	Intermittent	Elevated plasma ammonia concentration >100 to 150 micromol/L measured in arterial or venous blood sample. Plasma amino acid: ↑↑↑ Citrulline, ↑ glutamine ↑ urine orotic acid Neuroimaging during acute presentation may show cerebral edema, during prolonged hyperammonemia cortical atrophy, white matter cystic changes, hypomyelination Enzyme analysis: Fibroblasts from skin biopsy for ASS and ASL deficiencies DNA mutation analysis	Abnormal motor function, abnormal posturing, ataxia, spasticity	Frequent vomiting, nausea, poor appetite, food refusal, protein aversion, diarrhea, constipation, decreased level of consciousness, altered mental status, seizures, central hyperventilation, cerebral edema, progressive encephalopathy, intellectual disability, developmental delay	Protein-restricted diet, amino acid mixtures as needed, Nitrogen scavengers as needed Arginine-supplementation Emergency procedure: avoidance of catabolic states, anti-catabolic treatment (i.v. glucose) [7]	Outcome depends on the severity of the enzyme defect, the extent and duration of initial and recurrent hyperammonemia, as well as diagnostic delay
Galactosemia, classical (ORPHA:79239, OMIM: 230400) and Clinical variant galactosemia, combinded P: approximated 1/40,000–1/60,000	*GALT*, 9p13, Galactose-1-phosphate uridylyltransferase. Autosomal recessive	Chronic progressive	NBS in some countries ↑↑ galactose-1-phosphate in erythrocytes, ↑ free galactose in plasma and urine In untreated patients—elevated blood galactose urine reducing substances (galactosuria), Galactose-1-phosphate uridyltransferase deficiency in erythrocytes In affected neonates: liver failure, renal disease (hyperchloremic metabolic acidosis, aminoaciduria, elevated liver enzymes, albuminuria), (*E. coli*) sepsis	Ataxia, dystonia, tremor	On exposure to lactose: neonatal intoxication syndrome with hepatocerebral syndrome, vomiting, hypotonia, liver and renal failure, fulminant sepsis. Cognitive deficits, psychosocial challenges	Galactose-restricted diet for life	Improvement in clinical state, stabilisation of the disease Long-term complications occur despite diet
GLUT1 deficiency syndrome, (ORPHA:71277, *OMIM:* 606777) P: 1-9/1,000,000	*SLC2A1*, 1p34.2. Autosomal dominant or autosomal recessive	Intermittent	↓ glucose-ratio CRF/serum	Fluctuating (exacerbation during fasting) movement disorders, with ataxia, tremor, dystonia, chorea and/or spasticity	Severe form: Early-onset epileptic enephalopathy, secondary microcephaly, psychomotor retardation, intellectual disability; milder form: late-onset absence epilepsy	Ketogenic diet	Improved if early diagnosis
Hartnup disease, (ORPHA:2116, *OMIM:* 234500) P: 1-9/100,000	*SLC6A19*, 5p15.33, Neutral amino acid transporter. Autosomal recessive	Intermittent	Amino acid excretion patterns (Neutral hyperaminoaciduria)	Cerebellar ataxia, hyperreflexia, hypertonus	Seizures, developmental delay, emotional instability and psychosis, Light-sensitive dermatitis,	Supplementation with nicotinamide or tryptophan-rich diet; Patients on a high protein diet: sunlight protection and avoidance of photosensitizing drugs [11].	Reduced severity of symptoms
Hypobetalipoproteinemia (ORPHA:31154) P: unknown	*APOB*, 2p24, Apolipoprotein B. Autosomal recessive	Chronic progressive	↓ Serum cholesterol and beta lipoprotein	Ataxia, Ataxic movements of the hands, gait disturbance	Retinal degeneration, neuropathy, coagulopathy, hepatic steatosis, progressive demyelination of the central nervous system, lack of responsiveness to local anesthesia, dislike for animal fats and milk, psychomotor retardation, acantocytosis	Reduction in the proportion of fat in the patient’s diet and vitamin E supplementation in moderate form [11]	Prevention of adverse events
Isovaleric academia, (ORPHA:33, *OMIM:* 243500) P: 1-9/100,000	*IVD*, 15q14, isovaleric acid CoA dehydrogenase. Autosomal recessive	Intermittent	Newborn screening of plasma acylcarnitine profile (↑ C5), urine organic acids (↑↑ isovalerylglycine, OH-isovaleric acid)	Ataxia, other movement disorders	Developmental delay, cognitive impairment, aversion to dietary protein, episodes of massive metabolic acidosis in infancy (lethargy, vomiting, dehydration, impaired consciousness, seizures, odor of ‘sweaty feet’) moderate hepatomegaly, depressed platelets and leukocytes	Low protein diet, administration of glycine or l-carnitine, restriction of leucine intake	Prevention of further complications
Lesch-Nyhan disease, (ORPHA:510, *OMIM:* 300322) P: 1-9/1,000,000	*HGPRT1*, Xq26, Hypoxanthine-guanine phosphoribosyl-transferase. X-linked recessive	Chronic progressive	Prenatal diagnosis autoradiographic test for HPRT activity applied to cells obtained by amniocentesis	Dystonic movement disorder, and ataxia, spastic cerebral palsy, choreoathetosis	Compulsive self-injury behaviour with self-mutilation, Intellectual disability, dysarthria, ocular motor apraxia, impaired smooth pursuit, increased risk for nephrolithiasis, renal insufficiency, crystalluria, gout with tophi	Protective equipment (restraints), dental management. Atypical antipsychotics, e.g., risperidone, antidepressants, carbamazepine, gabapentin. 20S-adenosylmethionine (SAMe), a physiological intermediate in methylation and transsulfuration is beneficial in selected patients. DBS to reduce self-injury and/or modify dystonia	Reduction in self-injury and aggression, pharmacological management focuses on mood and anxiety stabilization, modification of dystonia
Maple syrup urine disease (ORPHA:268145, *OMIM:* 248600) P: 1-9/1,000,000	*BCKDHA*, 19q13.2, *BCKDHB*, 6q14.1, *DBT*, 1p21.2 or *DLD*, 7q31.1, Branched-chain amino acids-dehydrogenase. Autosomal recessive	Intermittent or Chronic progressive	Elevations of the branched-chain amino acids (BCAA) in plasma, α-ketoacids in urine Alloisoleucine in plasma	Characteristic “bicycling” movements in neonates, dystonia, spasticity, choreoathethosis, ataxia	Behavioural changes, nausea, vomiting, coma (attributed to cerebral edema), cognitive impairment, hyperactivity, sleep disturbance, hallucinations, up-gaze supranuclear palsy and adduction deficit, absence of voluntary eve movements, absent VOR	Low-protein die Leucine- free, isoleucine-, and valine-controlled amino acid supplements. Thiamine supplementation in thiamine-responsive MSUD Phenylbutyrate in MSUD Emergency treatment during metabolic stress, such as intercurrent illnesses	Improved outcome with timely introduction of specific diet treatment. Prevention of decompensation
Metachromatic leukodystrophy (ORPHA:512) P: 1-9/1,000,000	*ARSA*, 22q13, Arylsulfatase A. Autosomal recessive	Chronic progressive	Reduced nerve conduction velocity Elevated cerebrospinal fluid protein Genetics: pathogenic *ARSA* variant, Deficient arylsulfatase A (ARSA) enzyme activity in leukocytes. In patients with MLD, ARSA activity levels typically range from undetectable to less than 10 percent of normal values. CAVE: ARSA pseudodeficiency in ~1% of general population	Rigidity, muscle weakness and unsteady gait, ataxia, spasticity	Mental deterioration, convulsion, megacolon, hypotonia, schizophrenia, gallbladder nonfunctional, progressive apathy, memory disturbances, neuropathy	Bone marrow transplant, umbilical blood transplantation ERT, Clinical trials with lentiviral gene transfer	Continued developmental progress, improvement in neurophysiologic function and sulfatide metabolism New problems arise with treated phenotype
Mitochondrial disorders, including MERRF (P: unknown), MELAS (P: 1-9/1,000,000), NARP (P: 1-9/100,000), MILS (P: unknown), KSS	*mtDNA*, *nuclear genes* mitochondrial DNA (heteroplasmia), but also nuclear DNA variants	Intermittent, non-progressive, chronic progressive	↑ lactic acid (+/− pyruvat and other related metabolites of energy metabolism, including Krebs cycle) in plasma, urine, CSF and MR spectroscopy Paradoxical postprandial ketosis and other markers Abnormal OXPHOS/Oxymetric analyses in fibroblasts (or other cells, including urothelium) and muscle (or liver, kidney) biopsy histology	Any movement disorder, in any combination	Any organ may be affected: mental retardation, epilepsy, myopathy, cardiomyopathy, conduction defect (e.g., WPW syndrome), nephropathy, liver disease, retinitis pigmentosa, diabetes, exocrine pancreas insufficiency	Trial with co-factor substitution, e.g., thiamine, riboflavine, coenzyme Q10, carnitine, vitamine C, vitamine E, etc. [12,13]	apart from specific defects, little evidence for efficacy of any specific intervention
Multiple carboxylate deficiency (ORPHA:148), Biotin responsive and Biotidinase deficiency, P: 1-9/100,000	*HLCS,* 21q22, Holocarboxylase synthetase *BTD*, 3p25, leads to secondary deficiency of: Pyruvate carboxylase Propionyl-CoA carboxylase 3-methylcrotonyl-CoA carboxylase Acetyl-CoA carboxylase. Autosomal recessive	Intermittent or chronic progressive	NBS in some countries Biotinidase activity measured in plasma, (or in fibroblasts and leukocytes and other tissue extracts by radioassay [14].) Untreated patients: metabolic ketoacidosis, lactic acidosis, hyperammonemia; Elevation of 3-hydroxyisovaleric, 3-hydroxypropionic, lactic acid, and 3-methylcrotonylglycine in urine organic acid analysis. Urinary excretion of 3-hydroxyisovaleric acid- an indicator of biotin status.	Ataxia,	Ketoacidosis, lactic acidosis, dermatitis, alopecia, severe hypotonia, developmental regression, seizures, infantile spasms, infections, atrophy of the superior vermis of the cerebellum, hearing loss, optic atrophy Age at manifestation, for MCD usually neonatal, for biotinidase deficiency usually in childhood, or any time	Oral supplementation with biotin 5–20 (−40) mg/day (higher doses needed in HLCS deficiency)	Improvement or prevention with pharmacological doses of oral biotin [15]
Niemann-Pick type A and B (ASMD = acid sphingomyelinase deficiency) (ORPHA:618899) P: unknown	*SMPD1*, 11p15.4-p15.1, Acid sphingomyelinase. Autosomal recessive	Chronic progressive	↑ oxysterols (as in NPC, somewhat less consistently) Elevated serum triglycerides and LDL-cholesterol Low HDL-cholesterol ASM enzyme activity	Spasticity, rigidity, ataxia	Neurodevelopmental delay Type A: failure to thrive in neonatal period, hypotonia, failure to reach developmental milestones, rarely survival beyond 2–3 years Type B: little neurological involvement, pulmonary disease, frequent infections, hepatosplenomegaly, frequently living into adulthood	Treatment with olipudase alfa i.v. up to 3.0 mg/kg	Improvements in disease pathology
Niemann-Pick type C, (ORPHA:646) P: 1-9/100,000	*NPC1*: 18q11-q12, NPC1 protein *NPC2*: 14q24.3, NPC2 protein. Autosomal recessive	Chronic progressive	↑↑ Oxysterols Plasma bile acids: 3β,5α,6β-trihydroxycholanic acid, lysosphingolipids e.g., lyso-SM-509 Filipin staining Vertical supranuclear saccade and gaze palsy	Early-late infancy cases: Cataplexy Juvenile and adolescent cases: ataxia	Early-late infancy cases: Regression of already gained neurological skills in infancy, epilepsy, vertical supranuclear saccade, gaze palsy With mutations in the *NPC2* gene often respiratory insufficiency Juvenile and adolescent cases: dysarthrophonia, cognitive decline, psychiatric involvement, vertical supranuclear saccade, gaze palsy, epilepsy	Treatment with miglustat 200 mg tid p.o. in patients > 12 years, or weight-tiered dosis in children < 12 years [16] Side effects: diarrhoea (lack of selectivity affecting intestinal disaccharidases versus GCS and GbA2, tremor. Not authorities-approved yet: Acetyl-L-Leucine (ALL): Adult patients aged ≥13 years is 4 g per day (2 g in the morning, 1 g in the afternoon, and 1 g in the evening). Pediatric patients, the weight-tiered dosage; patients between 15 to <25 kg 2 g per day (1 g in the morning and 1 g in the evening); patients between 25 to <35 kg 3 g per day (1 g in the morning, 1 g in the afternoon, and 1 g in the evening), and patients > 35 kg 4 g per day at the same interval as adults	*Miglustat:* Stabilisation of disease, reduction in the progression rate [17] *ALL:* Improvement in signs and symptoms, reduction in disease progression [18,19]
Ornithine transcarbamylase deficiency, (ORPHA:664, *OMIM:* 311250) P: 1-9/100,000	*OTC*, Xp21.1, Ornithine transcarbamylase. X-linked inheritance. Penetrance of disease in females depends on X-inactivation, generally less severe than in (hemizygous) males.	Intermittent	Elevated plasma ammonia concentration >100 to 150 micromol/L measured in arterial or venous blood sample. Newborn screening in some countries Plasma amino acid/urine orotic acid analyses to differentiate among UCDs: ↑ glutamine, ↑ alanine, ↓citrulline, ↓ arginine, in plasma; ↑↑ orotic acid in urine Neuroimaging during acute presentation may show cerebral edema, during prolonged hyperammonemia cortical atrophy, white matter cystic changes, hypomyelination Enzyme analysis: in liver (and intestinal) biopsies	Abnormal motor function, abnormal posturing, ataxia, spasticity	Frequent vomiting, nausea, poor appetite, food refusal, protein aversion, diarrhea, constipation, decreased level of consciousness, altered mental status, seizures, central hyperventilation, cerebral edema, progressive encephalopathy, intellectual disability, developmental delay	Protein-restricted diet, amino acid mixtures as needed, Nitrogen scavengers as needed Arginine- (or citrulline-) supplementation Emergency procedure: avoidance of catabolic states, anti-catabolic treatment (i.v. glucose) [7,20]	Outcome depends on the severity of the enzyme defect, the extent and duration of initial and recurrent hyperammonemia, as well as diagnostic delay
Pyruvate dehydrogenase complex/pyruvate deficiency, (ORPHA:79243, *OMIM:* 312170) P: <1/1,000,000	*PDHA1*, Xp22.2-p22.1, E1-alpha subunit of the PDH enzyme complex. X-linked dominant	Intermittent	Lactic acid & pyruvate ↑ in plasma Postprandial increase in lactate & pyruvate, while normalization of lactate/pyruvate-ratio urinary organic acids Brain MRI Enzyme analysis of skin fibroblasts, Western blots, etc.	Ataxia, spasticity	Seizures, impaired cognition	Ketogenic diet Dichloroacetate or in some cases phenylbutyrate Thiamine, carnitine or lipoic acid supplementation	Significant improvement in symptoms
Refsum disease, (ORPHA:773, *OMIM*: 266500) P: 1-9/1,000,000	*PHYH*, 10p13, Phytanoyl CoA hydroxylase. Autosomal recessive	Chronic progressive	↑ Serum phytanic acid Phytanic acid oxidase activity in fibroblasts	Ataxia, limb paresis or atrophy	Neuropathy, retinitis pigmentosa with night blindness and constriction of the visual fields, cataracts, skin changes, polyneuropathy, impaired reflexes, sensory disturbances, anosmia, progressive hearing loss, cardiac abnormalities	Diet reduced in phytol and phytanic acids Plasmapheresis	Prevent or reverse symptoms
Sandhoff disease (GM2-Gangliosidosis), (ORPHA:796, *OMIM:* 268800) P: 1-9/1,000,000	*HEXB*, 15q23-q24, Alpha subunit of hexosaminidase A and B. Autosomal recessive	Chronic progressive	Chromatography of oligosaccharides (screening) Enzymatic essay of β-hexosaminidase activities carried out in plasma and/or peripheral white blood cells	Ataxia with involvement of 2nd motoneuron	Slowly developing intellectual disability, bipolar disorder, psychosis	No specific treatment, muscle-relaxing agents e. g. baclofen or tizanidine ALL improves motor signs and symptoms [21], not yet approved	Chronic progressive at different rates
Tay–Sachs disease (GM2-Gangliosidosis), (ORPHA:845, *OMIM:* 272800) P: unknown	*HEXA*, 15q23-q24, Alpha subunit of hexosaminidase A. Autosomal recessive	Chronic progressive	Chromatography of oligosaccharides (screening) Enzymatic essay of β-hexosaminidase activities carried out in plasma and/or peripheral white blood cells	Ataxia with involvement of 2nd motoneuron	Slowly developing intellectual disability, bipolar disorder, psychosis	No specific treatment, Baclofen, etc. ALL improves motor signs and symptoms [21], not yet approved	Chronic progressive at different rates
Wilson disease, (ORPHA:905, *OMIM:* 277900) P: 1-9/100,000	*ATPB7*, 13q14, ATP7B-Copper transporting ATPase. Autosomal recessive	Chronic progressive	Ceruloplasmin isoforms measurement in cord blood sample of neonates or venous blood sample of adults Urinary copper excretion levels Neuroophthalmologic slit-lamp assessment Neuroimaging	Ataxia, spasticity, dystonia, parkinsonism, tremor	Manifestations of liver cirrhosis	Chelation agents e. g. penicillamine, trientine, reduction in intestinal copper uptake with zinc, Triheptanoin	Prevention of neuronal damage, preservation of liver function

## Data Availability

This is a review article.

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
