# Peer review of "Inborn Errors of Metabolism with Ataxia: Current and Future Treatment Options"

_cells, 2023, doi:10.3390/cells12182314_

Round 1

Reviewer 1 Report

This is a well-written, comprehensive review on hereditary ataxias that are caused by inborn errors of metabolism (IEM), written mainly from a clinical perspective. In order to further improve it, a few important suggestions concerning mainly the presentation and clarity of the text are provided below:

1) For clarity, the presented table should utilize the Orphanet database identifiers (Orpha IDs). Similarly, the OMIM identifiers will be of use here. Please provide them in the table and throughout the text.

2) The table shall also provide the information about the prevalence, as majority of the disorders are rare (very rare),

3) The names of human genes are written in capital letters (in italic), while of proteins without. Please harmonize the usage of these throughout the whole text. Please also spell out the underlying names of the proteins using UniProtKB/SwissProt database.

4) It would be of potential interest to wider audience of readers if some information about the possible molecular targets of used drugs will be provided. Please revise the text accordingly.

The authors should recheck the manuscript in terms of usage of terms, double spacing, commas and phrasing. Some issues were spotted throughout the manuscript, and in the presented Table 1.

Please rephrase the following sentences:

1) "In some diseases, the first efforts of gene therapies in experimental animal studies have been done5". (line 67),

2) Table 1; "Late-adult on-86 sets, in persons older than 60 are possible".

3) "plasmaammonia" shall be "plasma ammonia",

4) "creatin peak", shall be "creatine peak". Similarly, "Acetyl-L-Leucin" should be "Acetyl-L-Leucine".

5) Please rephrase: "Phenylbutyrate acts on the kinase of the BCKAD and seems becoming an additional treatment option in MSUD13 (see also NCT01529060).

6) "Ex vivo" shall be written in italic,

7) Please rephrase: "It is an iminosugar that crosses the BBB), acting primarily as substrate reduction."

Reviewer 2 Report

I would like to star by congratulating the authors on their interesting article. They provide us with a comprehensive review on ataxias caused by inborn errors of metabolism, with a particular effort on its treatment.

The article is little too long and for that, it might be difficult to read as a whole. As the topic of Cells issue is on treatment, I would suggest the authors to shorten on the descriptive part of the diseases. Also, they should try to have more or less the same amount of space into disease’s description, always highlighting the ataxia presentation and then the associated features which may point into that diagnosis.

Treatments description should be more homogenous. Some treatments, even if only in clinical trials phase, are too extended, while others are much reduced. For instance, on NPC type C there’s little information on miglustat and two large paragraphs on ALL. It should be preferred to describe in more detail the mode of action and results from trials of already approved drugs.

I would suggest for the table to be on landscape view, and repeat the headlines in every page.

Minor grammar misspellings, with the use of capital letters in the middle of the text or low case letters in the beginning of some sentences on table.

Round 2

Reviewer 1 Report

The authors have sufficiently answered and followed the Reviewer's recommendations. I wish to congratulate the authors on bringing this interesting and important topic to the wider audience of readers.